# *Salmonella enterica* serovar Typhimurium chitinases modulate the intestinal glycome and promote small intestinal invasion

Jason R. Devlin[1], William Santus[1], Jorge Mendez[1], Wenjing Peng[2], Aiying Yu[2], Junyao Wang[2], Xiomarie Alejandro-Navarreto[1], Kaitlyn Kiernan[1], Manmeet Singh[3], Peilin Jiang[2], Yehia Mechref[2], Judith Behnsen[1] *

1 Department of Microbiology and Immunology, University of Illinois Chicago, Chicago, Illinois, United States of America, 2 Department of Chemistry and Biochemistry, Texas Tech University, Lubbock, Texas, United States of America, 3 Department of Pathology, University of Illinois Chicago, Chicago, Illinois, United States of America

* jbehnsen@uic.edu

**Data Availability Statement:** Glycome analysis data set is available in the GlycoPOST repository via the accession number GPST000225. All other relevant data is present within this manuscript.

## Abstract

*Salmonella enterica* serovar Typhimurium (*S.* Typhimurium) is one of the leading causes of food-borne illnesses worldwide. To colonize the gastrointestinal tract, *S.* Typhimurium produces multiple virulence factors that facilitate cellular invasion. Chitinases have been recently emerging as virulence factors for various pathogenic bacterial species, and the *S.* Typhimurium genome contains two annotated chitinases: *STM0018* (*chiA*) and *STM0233*. However, the role of these chitinases during *S.* Typhimurium pathogenesis is unknown. The putative chitinase STM0233 has not been studied previously, and only limited data exists on ChiA. Chitinases typically hydrolyze chitin polymers, which are absent in vertebrates. However, *chiA* expression was detected in infection models and purified ChiA cleaved carbohydrate subunits present on mammalian surface glycoproteins, indicating a role during pathogenesis. Here, we demonstrate that expression of *chiA* and *STM0233* is upregulated in the mouse gut and that both chitinases facilitate epithelial cell adhesion and invasion. *S.* Typhimurium lacking both chitinases showed a 70% reduction in invasion of small intestinal epithelial cells *in vitro*. In a gastroenteritis mouse model, chitinase-deficient *S.* Typhimurium strains were also significantly attenuated in the invasion of small intestinal tissue. This reduced invasion resulted in significantly delayed *S.* Typhimurium dissemination to the spleen and the liver, but chitinases were not required for systemic survival. The invasion defect of the chitinase-deficient strain was rescued by the presence of wild-type *S.* Typhimurium, suggesting that chitinases are secreted. By analyzing *N*-linked glycans of small intestinal cells, we identified specific N-acetylglucosamine-containing glycans as potential extracellular targets of *S.* Typhimurium chitinases. This analysis also revealed a differential abundance of Lewis X/A-containing glycans that is likely a result of host cell modulation due to the detection of *S.* Typhimurium chitinases. Similar glycomic changes elicited by chitinase deficient strains indicate functional redundancy of the chitinases. Overall, our results demonstrate that *S.* Typhimurium chitinases contribute to intestinal adhesion and invasion through modulation of the host glycome.

**Funding:** This work was supported by the National Institutes of Health (1R01AI143641 to JB and 1R01GM112490 to YM) and the Department of Microbiology at the University of Illinois Chicago (to JB). The funders had no role in study design, data collection and analysis, decision to publish, or preparation of the manuscript.

**Competing interests:** The authors have declared that no competing interests exist.

## Author summary

*Salmonella* Typhimurium infection is one of the leading causes of food-borne illnesses worldwide. In order for *S.* Typhimurium to effectively cause disease, it has to invade the epithelial cells lining the intestinal tract. This invasion step allows *S.* Typhimurium to replicate efficiently, causing further tissue damage and inflammation. In susceptible patients, *S.* Typhimurium can spread past the intestines and infect peripheral organs. It is essential to fully understand the invasion mechanism used by *S.* Typhimurium to design better treatments for infection. Here, we demonstrate that the two chitinases produced by *S.* Typhimurium are involved in this invasion process. We show that *S.* Typhimurium chitinases interact with surface glycans of intestinal epithelial cells and promote adhesion and invasion. Using a mouse infection model, we show that *S.* Typhimurium chitinases are required for the invasion of the small intestine and enhance the dissemination of *S.* Typhimurium to other organs. This study reveals an additional mechanism by which *S.* Typhimurium invades and causes infection.

## Introduction

As a food-borne pathogen, *S.* Typhimurium colonizes the gastrointestinal tract causing self-limiting gastroenteritis. A critical step in the infection cycle of *S.* Typhimurium is the invasion of intestinal epithelial cells via the type-3 secretion system-1 (T3SS-1) [1]. Once inside, *S.* Typhimurium hyper-replicates and triggers pyroptosis of the intestinal epithelial cell, releasing newly formed bacterial cells back into the intestinal lumen, exacerbating infection [2]. Immuno-compromised individuals are more susceptible to infection due to the ability of *S.* Typhimurium to breach the epithelial barrier and disseminate to peripheral organs, causing a systemic infection [3]. *S.* Typhimurium expresses a variety of virulence factors that promote this pathogenic lifestyle. These provide functions ranging from aiding in the adhesion to and invasion of intestinal epithelial cells, promoting intracellular survival, and modulating the host immune response [4–6]. Many of these virulence factors have been studied in detail, while others are continuously being discovered. Two chitinases present in the genome of *S.* Typhimurium LT2 (NCBI: NC_003197.2), STM0018 (ChiA; GeneID:1251536) and STM0233 (GeneID: 1251751), have been recently proposed as potential virulence factors [7]. Our study uses the ATCC 14028 strain of *S.* Typhimurium (GenBank: CP001363.1), which has the genomic identifiers *STM14_0022* and *STM14_0275* for *chiA* and *STM0233*, respectively. For the sake of consistency, we will use the LT2 nomenclature to refer to these chitinases.

The main function of chitinases is the hydrolysis of chitin polymers into N-acetyl glucosamine (GlcNAc) oligomers [8]. Chitin is a component of the cuticle of insects and crustaceans and is a part of the fungal cell wall, making it the second most abundant biopolymer in nature. Despite chitin's prevalence, it is absent in mammalian species [8]. However, chitinases can also demonstrate catalytic activity towards other GlcNAc-containing polysaccharides, such as peptidoglycan [9]. This observation indicates that chitinases may interact with other biologically relevant polysaccharides to serve alternative roles. Interestingly, chitinases and chitin-binding proteins, which lack catalytic activity, are emerging as virulence factors for various pathogenic bacterial species. *Legionella pneumophila* expresses a chitinase that shows catalytic activity towards mucins and is required for colonization of the lungs [10,11]. The intestinal pathogen *Listeria monocytogenes* produces a chitinase that modulates inducible nitric oxide synthase (iNOS) expression to facilitate colonization of the liver and spleen [12,13]. *Vibrio cholerae*

produces a chitinase that degrades and allows the utilization of intestinal mucins as a carbon source [14]. *V. cholerae* also produces a chitin-binding protein that adheres to mucins to promote gastrointestinal colonization [14,15]. Both Adherent-Invasive *Escherichia coli* (AIEC) and *Serratia marcescens* produce a chitinase and a chitin-binding protein, respectively, that contribute to the adhesion to intestinal epithelial cells [16,17]. A mammalian chitin-binding protein, Chitinase 3 like 1, has already been implicated in *S.* Typhimurium infection by promoting the invasion of colonic epithelial cells *in vitro* and intestinal colonization and dissemination *in vivo* [18]. These previous studies indicate that the chitinases encoded by *S.* Typhimurium have the potential to contribute to human infection.

Despite the clear interest in the roles of chitinases during infection, *S.* Typhimurium chitinases have yet to be studied in detail, and their roles in gastrointestinal infection are still unknown. The putative chitinase STM0233 has not been studied experimentally, and its expression, enzymatic activity, and role during infection are entirely unknown. The expression of *chiA* has been detected during infection of epithelial cells, murine macrophages, and the gastrointestinal system of chickens [19–21]. The enzymatic activity of ChiA has been partially elucidated. As expected, ChiA shows *in vitro* hydrolytic activity towards chitin. However, ChiA also cleaves N-acetyllactosamine (LacNAc) [22,23], a common component of surface glycoproteins of intestinal epithelial cells. *S.* Typhimurium chitinases might therefore represent glycoside hydrolases that have flexible specificity and interact with the intestinal glycome. In many surface glycoproteins, LacNAc masks the underlying mannose subunits [24]. Exposing these mannose residues is potentially important for *S.* Typhimurium pathogenesis. An increase of high-mannose glycans was previously shown to increase the invasion of intestinal epithelial cells *in vitro* [25,26]. *S.* Typhimurium also produces a type 1 fimbria (FimH) that binds mannose subunits on surface glycoproteins to facilitate adhesion to host cells [27]. Therefore, we hypothesized that *S.* Typhimurium chitinases are remodeling the intestinal glycome by removing LaNAc to expose mannose residues that facilitate the adhesion to and invasion of intestinal epithelial cells. Here, we demonstrate that both *S.* Typhimurium chitinases are required to adhere to and invade intestinal epithelial cells *in vitro*. In a mouse model, chitinases facilitated small intestinal cell invasion, which contributed to *S.* Typhimurium dissemination to peripheral organs. The presence of *S.* Typhimurium chitinases resulted in specific changes in the abundance of GlcNAc-containing *N*-linked surface glycans, indicating that chitinases cleave these GlcNAc residues. *S.* Typhimurium chitinases also stimulated host cells to upregulate Lewis X/A-containing glycans and other complex glycan species.

## Results

### S. Typhimurium chitinases are required for intestinal epithelial cell adhesion and invasion

Since there are no experimental studies on STM0233 and only limited studies of ChiA in the current literature [22,23], we set out to determine the roles of these two chitinases for *S.* Typhimurium growth, interaction with host cells, and pathogenesis. We first confirmed that both chitinases are expressed by *S.* Typhimurium *in vitro*. The expression of *STM0233* and *chiA* mRNA was detectable by RT-qPCR. Wild-type (WT) *S.* Typhimurium expressed both chitinases under growth conditions relevant to our *in vitro* studies (LB broth and DMEM/F12 + 10% FBS) (Fig 1A). We observed a 2-fold upregulation of *STM0233* in DMEM/F12, but a similar expression of *chiA* across both media (Fig 1A). We generated single- and double-deletion strains lacking the chitinase encoding genes and confirmed that the strains have similar growth kinetics as WT in rich medium (S1A Fig). One potential role for *S.* Typhimurium chitinases is the degradation and utilization of dietary chitin as a carbon source. We thus

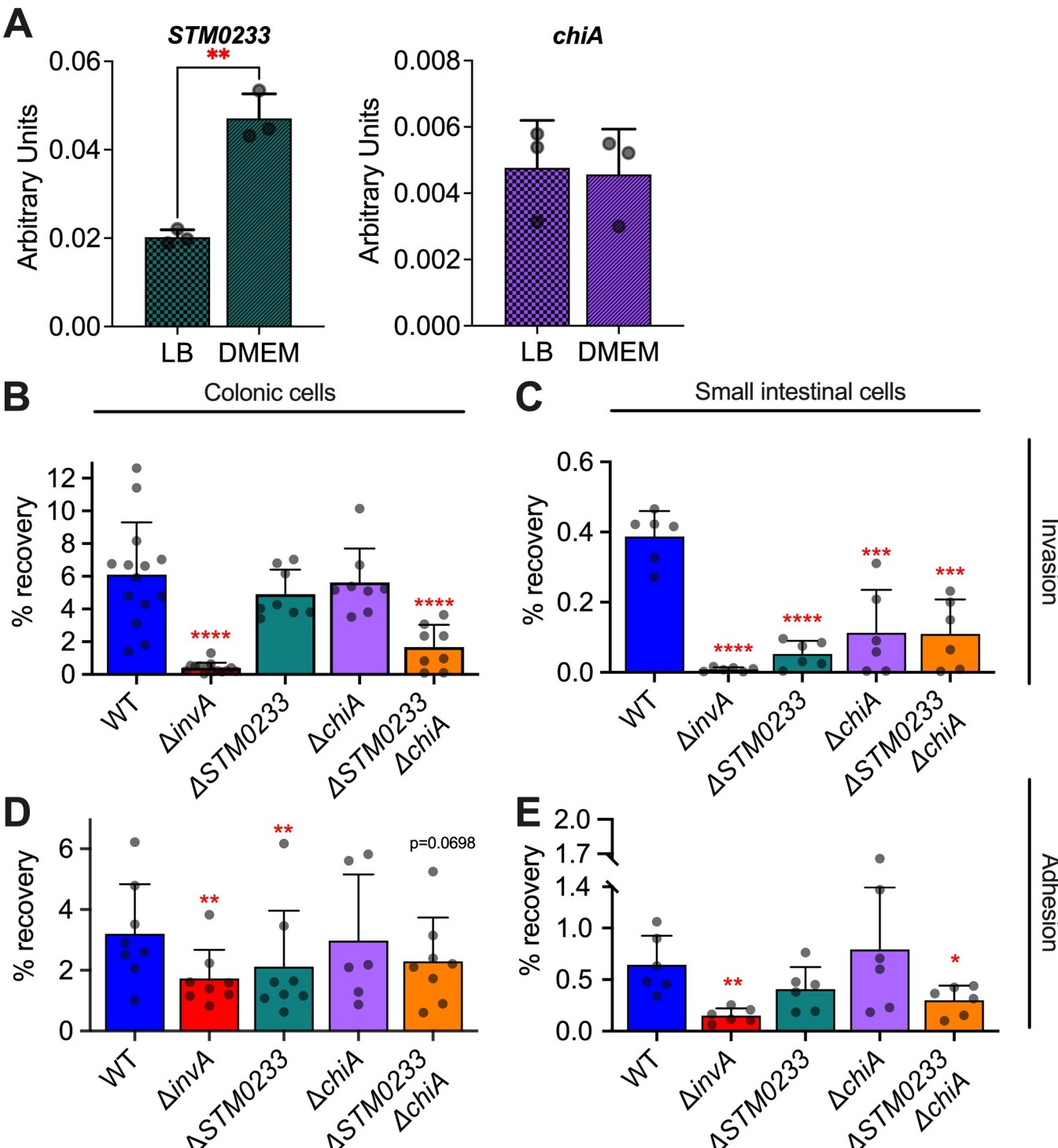

**Fig 1. *S.* Typhimurium chitinases contribute to intestinal epithelial cell adhesion and invasion *in vitro*.** (A) mRNA expression of *S.* Typhimurium chitinases during mid-log phase of growth in LB or DMEM/F12 + 10% FBS measured by RT-qPCR. Expression is normalized to the housekeeping gene *gmk*. n = 3. (B) Gentamicin protection assay of *S.* Typhimurium infected (MOI:1) colonic epithelial cells (T84). WT n = 14, Δ*invA* n = 14, Δ*STM0233* n = 8, Δ*chiA* n = 8, Δ*STM0233* Δ*chiA* n = 8. (C) Gentamicin protection assay of *S.* Typhimurium infected (MOI:1) small intestinal epithelial cells (IPEC-1). n = 6. (D) Adhesion assay of Cytochalasin D treated (2 μg/mL), *S.* Typhimurium infected (MOI:1) colonic epithelial cells (T84). WT n = 8, Δ*invA* n = 8, Δ*STM0233* n = 8, Δ*chiA* n = 6, Δ*STM0233* Δ*chiA* n = 8. (E) Adhesion assay of Cytochalasin D treated (2 μg/mL), *S.* Typhimurium infected (MOI:1) small intestinal epithelial cells (IPEC-1). n = 6. Bars represent mean ± SD. Statistics: (A) unpaired *t*-test. (B-E) Stars indicate significance compared to the WT control by (B, D) one-way ANOVA with Dunnett's multiple comparison test or (C, E) mixed-effect analysis with Dunnett's multiple comparison test. * = p<0.05, ** = p<0.01.

performed growth curves in minimal media supplemented with colloidal chitin but found no difference in the growth of strains lacking chitinases compared to WT (S1B Fig). The lack of a growth defect for the chitinase-deficient strains indicates that *S.* Typhimurium chitinases are not involved in utilizing chitin as a carbon source.

ChiA's activity towards LacNAc *in vitro* indicates that this chitinase has the potential to interact with surface glycoproteins of intestinal epithelial cells [22,23]. Interactions with surface glycoproteins have been previously shown to promote *S.* Typhimurium invasion [25–27]. We, therefore, hypothesized that *S.* Typhimurium chitinases facilitate the invasion of intestinal epithelial cells to promote infection. We assessed the capabilities of the chitinase-deficient strains to invade epithelial cells of the small intestine (IPEC-1) and colon (T-84) by performing a gentamicin protection assay. An *invA* (GeneID:1254419) deletion strain was used as a negative control, as it lacks expression of a functional T3SS-1, which is required to invade this cell type [28]. *S.* Typhimurium strains deficient in only one chitinase showed a trend towards reduced invasion of colonic cells, whereas the strain deficient in both chitinases (Δ*STM0233* Δ*chiA*) showed a highly significant >70% reduction in invasion compared to WT (Fig 1B). In the small intestinal epithelial cell line IPEC-1, invasion deficiencies of *S.* Typhimurium strains lacking chitinases were even more evident. Here, strains deficient in only one chitinase also showed a >70% reduction in invasion, similar to the double-chitinase deletion strain (Fig 1C). The complementation of the deleted chitinase genes restored invasion of the chitinase-deficient strains (S2 Fig). Therefore, both STM0233 and ChiA contribute to the invasion of intestinal epithelial cells, and deletion of either chitinase seems to have a greater effect on small intestinal invasion than colonic invasion.

Invasion of intestinal epithelial cells by *S.* Typhimurium requires successful adhesion and the deployment of the T3SS-1 [29,30]. We, therefore, investigated whether chitinases contribute directly to invasion or if their primary role is influencing adhesion. Intestinal epithelial cells were incubated with cytochalasin D prior to and during infection to inhibit the actin rearrangement required for *S.* Typhimurium invasion. Cytochalasin D treatment completely blocked the invasion of colonic and small intestinal cells (S3A and S3B Fig). It is currently thought that *S.* Typhimurium adhesion is mediated by the sequential, transient binding of fimbrial adhesins, followed by a more stable adhesion driven by the deployment of the T3SS-1 [31,32]. We used the Δ*invA* mutant as a negative control, as it is known to have an adhesion defect due to its lack of a functional T3SS-1 [31,32]. In colonic epithelial cells, the Δ*STM0233* strain showed a 33% reduction in adhesion compared to WT (Fig 1D), indicating a potentially larger role for STM0233 in adhesion than ChiA. The double-deletion strain also showed a trend toward reduced adhesion (p = 0.0698; Fig 1D). In the small intestinal cell line, the double-deletion strain showed a 50% reduction in adhesion (Fig 1E), similar to the observed 70% reduced invasion (Fig 1C). Although chitinases seem to play a partial role in contributing to colonic adhesion, both chitinases significantly contributed to the adhesion to small intestinal cells.

## *S.* Typhimurium chitinases are required for invasion and colonization of the small intestines *in vivo*

Given the significant role of chitinases as invasion factors for epithelial cells *in vitro*, we next explored if chitinases contribute to *S.* Typhimurium pathogenicity *in vivo*. We used a streptomycin-pretreatment mouse model (Fig 2A), in which C57BL/6 mice were treated with streptomycin 24 h prior to infection to promote *S.* Typhimurium colonization of the intestinal tract and to allow *S.* Typhimurium to trigger intestinal inflammation similar to human infection [33]. We first examined the expression of chitinase genes in colonic luminal samples collected

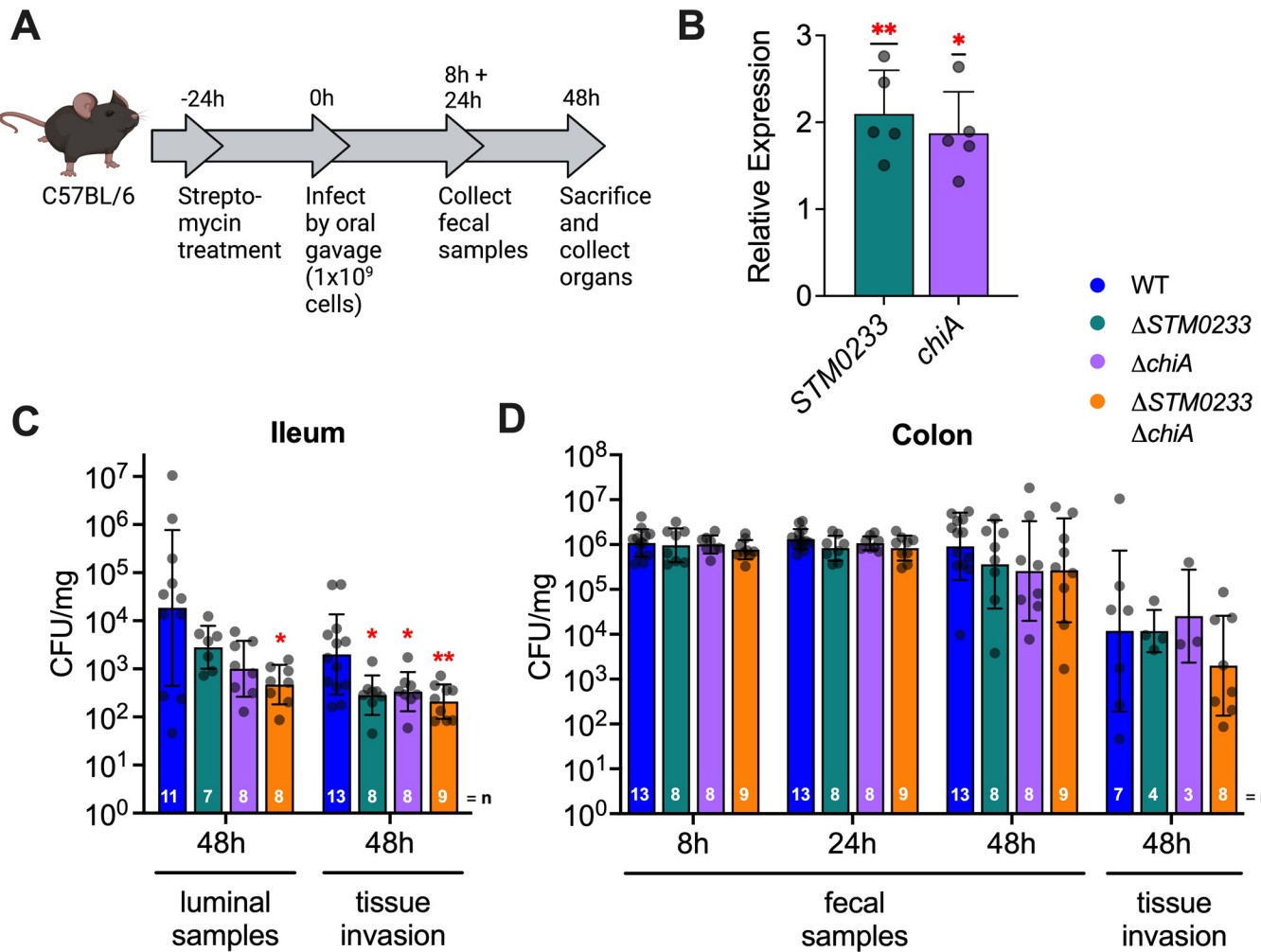

**Fig 2. *S.* Typhimurium chitinases are required for the colonization and invasion of small intestinal epithelial cells *in vivo*.** (A) Streptomycin pre-treatment mouse model of *S.* Typhimurium infection. (B) mRNA expression of *S.* Typhimurium chitinases extracted from colonic luminal samples collected 48 hpi. Expression is normalized to the housekeeping gene *gmk*. Expression is represented relative to mid-log expression in LB. n = 5. Bars represent mean ± SD. (C) Luminal samples from the terminal ileum were collected 48 hpi to determine *S.* Typhimurium colonization. Invasion was determined with a gentamicin protection assay performed on the terminal ileum. (D) *S.* Typhimurium colonies recovered from fecal samples collected at 8 and 24 hpi. Fecal samples at 48 hpi were collected directly from the lumen of the colon. Invasion of colonic tissue was determined with a gentamicin protection assay. Bars represent geometric mean ± geometric SD. Statistics: (B) one-sample *t*-test against our null hypothesis (relative expression = 1) (C-D) Stars indicate significance compared to the WT control by one-way ANOVA with Dunnett's multiple comparison test. * = $p < 0.05$, ** = $p < 0.01$.

48 hours post-infection (hpi). The expression of both *STM0233* and *chiA* was significantly upregulated *in vivo* compared to LB broth (Fig 2B). We further examined *S.* Typhimurium invasion of the ileum and colon by performing a gentamicin protection assay on intestinal tissue and determined luminal colonization levels. The chitinase-deficient strains displayed markedly reduced invasion of ileal tissue at 48 hpi, with the double-deletion strain showing a 10-fold invasion defect compared to *S.* Typhimurium WT (Fig 2C). Simultaneously, chitinase-deficient strains also showed a defect in the colonization of the lumen of the ileum (Fig 2C). Surprisingly, there was no defect in the ability of the chitinase-deficient strains to invade colonic tissue or colonize the colon at 48 hpi (Fig 2D). Consistent with this finding, chitinase-deficient strains did not show decreased colonization in fecal samples collected throughout infection (Fig 2D). Therefore the *in vivo* invasion defect is specific to the small intestines,

despite the invasion defect observed during *in vitro* infection of colonic epithelial cells (Fig 1B).

## *S.* Typhimurium chitinases are required for dissemination during the early stages of gastrointestinal infection but are dispensable during systemic infection

Early during infection, *S.* Typhimurium specifically targets M cells and invades Peyer's patches [34]. *S.* Typhimurium is then transported to the mesenteric lymph nodes by antigen-presenting cells and eventually colonizes the spleen and liver, entering systemic circulation [35]. We, therefore, investigated the ability of the chitinase-deficient strains to disseminate to peripheral organs during infection. Both the *chiA* (14-fold) and the double-deletion strain (50-fold) showed significantly reduced colonization of the spleen at 48 hpi compared to WT *S.* Typhimurium (Fig 3A). Strikingly, for all but one mouse, the double-chitinase deletion strain did not colonize the liver to detectable levels, in contrast to WT *S.* Typhimurium (Fig 3A). This colonization defect may be due to a direct role of chitinases for *S.* Typhimurium survival at systemic sites. To rule out this possibility, we administered *S.* Typhimurium directly into the peritoneum. Without the requirement of intestinal invasion, chitinase-deficient strains were able to colonize the spleen and the liver to a similar extent as WT at both 24 and 48 hpi (Fig 3B). These results indicate that chitinase-mediated invasion of the intestines leads to an increase in dissemination. No colonization defect was detected in the Peyer's patches or mesenteric lymph nodes at 48 hours post-gastrointestinal infection (Fig 3A). These sites are the first that *S.* Typhimurium disseminates to during infection [35]. An initial defect in the colonization of the Peyer's patches or mesenteric lymph nodes due to lower gastrointestinal invasion might thus be masked by bacterial replication.

The hypothesis that chitinases are mediators during the early stages of infection is supported by data from an extended infection mouse model (S4A Fig). When mice were infected for 96 h, both WT and chitinase-deficient strains showed similar invasion and colonization of the ileum (S4B Fig). Chitinase-deficient strains were also not defective in their ability to colonize the colon or disseminate to the Peyer's patches, mesenteric lymph nodes, spleen, or liver (S4C and S4D Fig). Our findings thus far show that chitinases are required during early infection but are dispensable for survival at systemic sites and during later stages of infection once *S.* Typhimurium has fully established colonization.

## *S.* Typhimurium chitinases are not involved in modulating the innate immune response

Based on the ability of the chitinase produced by *Listeria monocytogenes*, ChiA, to downregulate the expression of inducible nitric oxide synthase (iNOS) as a mechanism to promote colonization of the liver and spleen [12], we explored if *S.* Typhimurium chitinases interact with the immune system in a similar manner. We collected cecal tissue from mice infected with WT or chitinase-deficient strains for 48h and analyzed the expression of a panel of genes that are known to be involved in the immune response to *S.* Typhimurium infection [6,36]. We found no changes in the expression of *Nos2* (MGI:97361), *Duox2* (MGI:3036280), *Cxcl1* (MGI:108068), *Ifng* (MGI:107656), *Il6* (MGI:96559), *Il22* (MGI:1355307), *Il23* (MGI:1932410), *S100a9* (encoding a subunit of the antimicrobial protein calprotectin; MGI:1338947), *Lcn2* (MGI:96757), *Il1b* (MGI:96543), or *Tnf* (MGI:104798) in mice infected with chitinase-deficient strains compared to WT infected mice (Fig 4A–4K). We detected a 2-fold upregulation of *Il17a* (MGI:107364) in mice infected with chitinase-deficient strains (Fig 4L). The biological relevance of this differential expression is unclear, as there are no changes in upstream

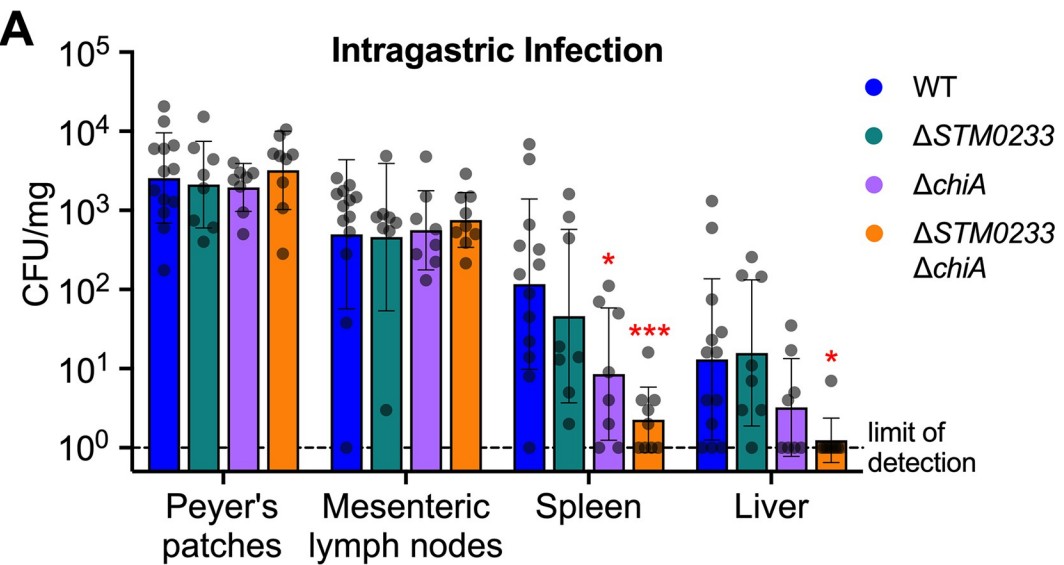

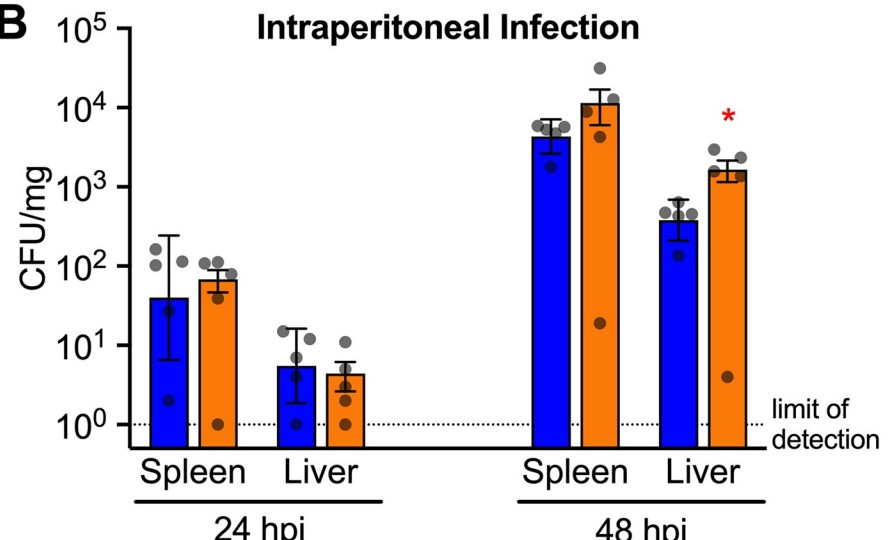

**Fig 3. Chitinase-mediated invasion of intestinal tissue contributes to *S.* Typhimurium dissemination.** (A) Colonization of the Peyer's patches, mesenteric lymph nodes, spleen, and liver 48 h after intragastric infection. WT n = 13, Δ*STM0233* n = 8, Δ*chiA* n = 8, Δ*STM0233* Δ*chiA* n = 9. (B) Colonization of the spleen and liver 24 h and 48 h after intraperitoneal infection. n = 5. Bars represent geometric mean ± geometric SD. Statistics: Stars indicate significance compared to the WT control by one-way ANOVA with Dunnett's multiple comparison test. * = p<0.05, *** = p<0.001.

regulators (*Il23*) or any downstream effectors (*Lcn2*, *Cxcl1*, *S100a9*). We additionally examined the expression of the innate immune genes *Cxcl1*, *Il6*, *Nos2*, *Il23a*, and *Il17a* in infected ileal tissue. Here, *Cxcl1*, *Il6*, *Il23a, and Il17a* trended towards higher expression (2–3 fold) in chitinase-deficient *S.* Typhimurium infected mice, but the difference was not statistically significant (S6A and S6B, S6D, and S6E Fig). *Nos2* expression remained similar across all infection groups (S6C Fig). Furthermore, histopathological analysis of small intestinal and cecal tissue showed no differences between mice infected with WT *S.* Typhimurium and mice infected with chitinase-deficient strains (Figs 4M and 4N, and S5A–S5D). Therefore, the small differences in gene expression do not result in a significant effect on overall inflammation. Given

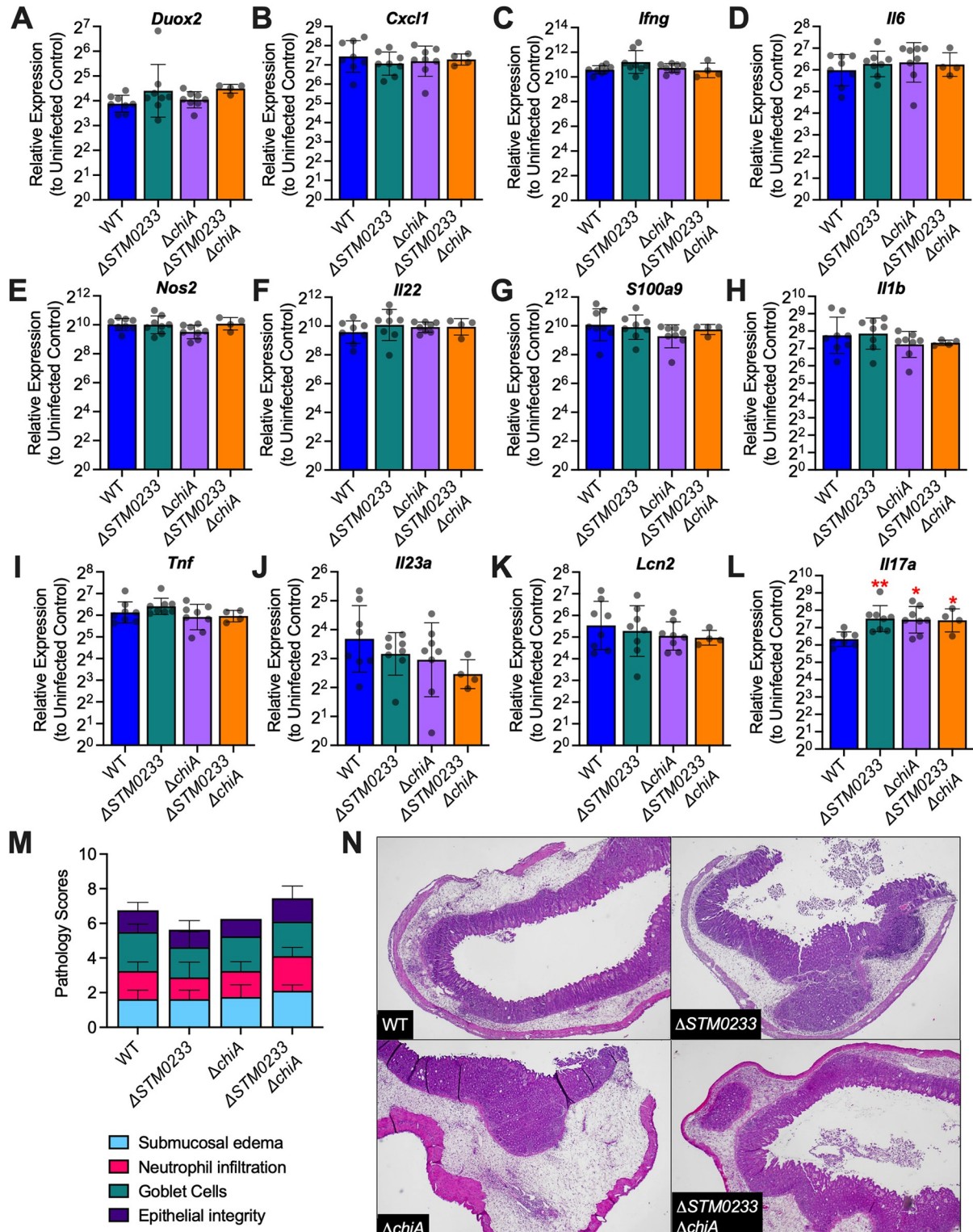

**Fig 4. *S*. Typhimurium chitinases do not modulate the innate immune response *in vivo*.** (A-L) mRNA expression of innate immune genes extracted from cecal tissue of *S*. Typhimurium infected mice 48 hpi. Expression is normalized to the housekeeping gene *actb* as well as gene expression in uninfected mice. WT n = 8, Δ*STM0233* n = 8, Δ*chiA* n = 8, Δ*STM0233* Δ*chiA* n = 4. Bars represent geometric mean ± geometric SD. (M) Histopathological scoring of cecal tissue from *S*. Typhimurium infected mice 48 hpi. WT n = 8, Δ*STM0233* n = 8, Δ*chiA* n = 8, Δ*STM0233* Δ*chiA* n = 9. Tissues were scored for submucosal edema, neutrophil infiltration, goblet cells, and epithelial integrity. Bars

represent mean ± SD. (N) Representative images of hematoxylin & eosin stained cecal tissue from *S.* Typhimurium infected mice 48 hpi. 400x magnification was used for WT and Δ*STM0233* Δ*chiA* images, 200x magnification was used for Δ*chiA* and Δ*STM0233* images. Statistics: (A-M) Stars indicate significance compared to the WT control by one-way ANOVA with Dunnett's multiple comparison test. * = p<0.05, ** = p<0.01.

the similarity in immune gene expression and the lack of differential pathological scores, we concluded that the phenotypes observed with chitinase-deficient strains were not driven by an altered immune response.

## The presence of wild-type chitinases can rescue mutant colonization and invasion

The chitinases ChiA and STM0233 are predicted to be secreted based on their amino acid sequences [37,38]. We, therefore, hypothesized that the chitinase-deficient strains would be able to utilize WT chitinases to enhance invasion if they are available during infection. We used the streptomycin-pretreatment mouse model to explore the invasive capabilities of the chitinase-deficient *S.* Typhimurium strain during co-infection with WT *S.* Typhimurium. After 48 h of co-infection, both strains invaded the colon or ileum at similar levels (Fig 5A and 5B). WT *S.* Typhimurium still showed higher colonization in the lumen of the ileum (Fig 5B). However, the competitive advantage for WT (2-fold) was drastically reduced compared to the difference in colonization observed during single-infection (39-fold) (Fig 2C). Analyzing organ dissemination also did not reveal a competitive advantage for WT *S.* Typhimurium (Fig 5C). These data demonstrate that the presence of WT *S.* Typhimurium can rescue the invasion and colonization defect of the chitinase-deficient strain. The secretion of chitinases would explain the lack of an invasion defect during co-infection, as the chitinase-deficient strain would still be able to utilize secreted WT chitinases.

## *S.* Typhimurium chitinases induce specific changes to the surface glycome of infected cells

To examine if *S.* Typhimurium chitinases directly interact with surface glycoproteins of small intestinal epithelial cells during infection, we analyzed the abundance of glycan species during *in vitro* infection with WT *S.* Typhimurium and the chitinase-deficient strains. Infection was performed at an MOI of 1000 to ensure that any glycomic changes would be detectable. We simultaneously performed an invasion assay to confirm that the chitinase-deficient strains also demonstrated the previously observed invasion defect at this higher MOI (S7 Fig). Fig 6A shows a common *N*-linked glycan species and its saccharide components. Principal component analysis (PCA) revealed distinct groupings for the glycome of uninfected, WT infected, and the chitinase-deficient strain infected cells, indicating that *S.* Typhimurium chitinases induce specific glycomic changes during infection (Fig 6B). Closer inspection of the changes in the relative abundance of glycans revealed an increase in the abundance of specific glycans when epithelial cells were challenged with the chitinase-deficient strains compared to WT infection. These included various glycans containing GlcNAc, LacNAc, or NeuAc-LacNAc as terminal residues (43100, 45100, 33000, 44000, 34010, 55020) (Fig 6C). We, therefore, hypothesized that *S.* Typhimurium chitinases might indeed cleave GlcNAc-containing residues on these glycans. However, other GlcNAc-containing glycans were unchanged in relative abundance when chitinases were deleted (S4 Table). *S.* Typhimurium chitinases may, therefore, be interacting with specific glycan species instead of broadly interacting with all GlcNAc-containing glycans. Surprisingly, ChiA and STM0233 seem to share similar activity towards specific

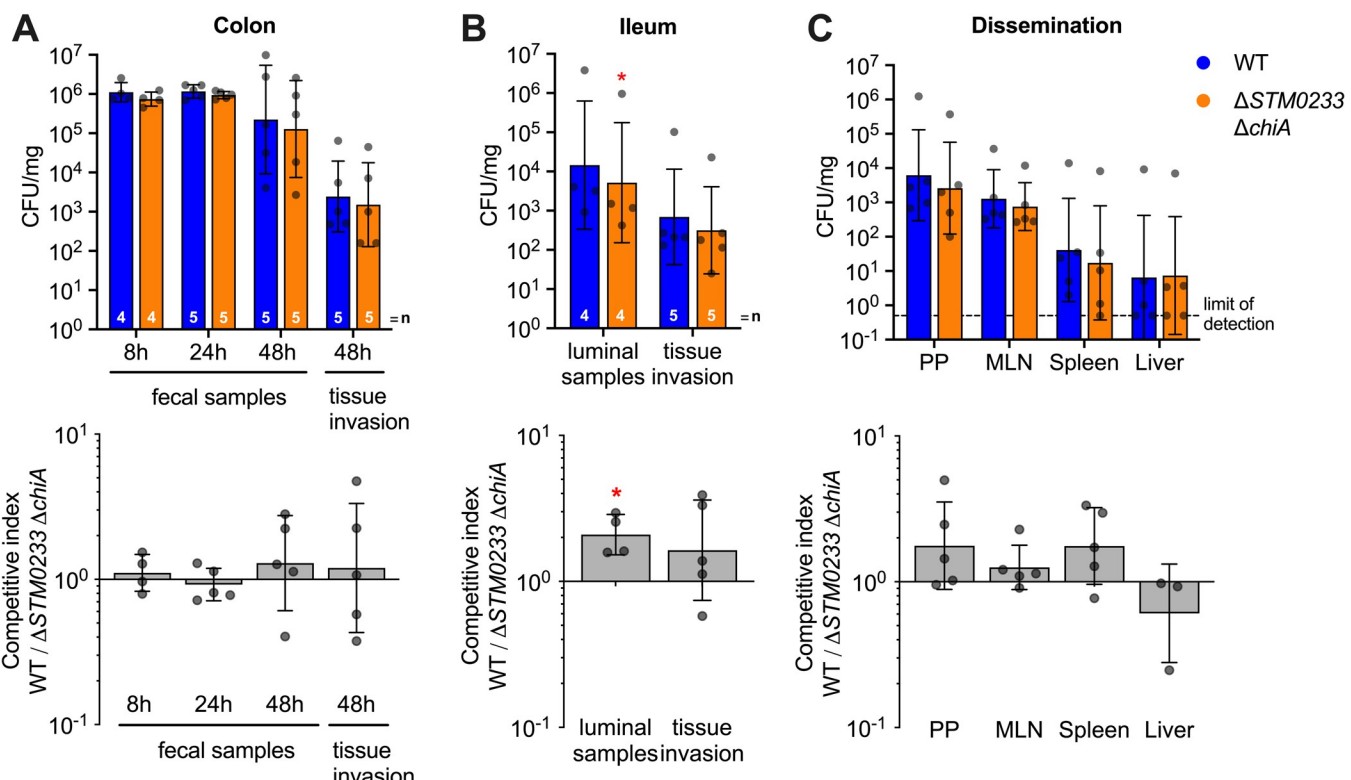

**Fig 5. The presence of WT *S*. Typhimurium rescues the invasion of chitinase-deficient *S*. Typhimurium.** Mice infected by oral gavage with a 1:1 ratio of WT and chitinase-deficient *S*. Typhimurium. (A) *S*. Typhimurium colonies recovered from fecal samples collected at 8 and 24 hpi. Fecal samples at 48 hpi were collected directly from the lumen of the colon. Invasion of colonic tissue was determined with a gentamicin protection assay. (B) Luminal samples from the terminal ileum were collected 48 hpi to determine *S*. Typhimurium colonization. Invasion was determined with a gentamicin protection assay performed on the terminal ileum. (C) Colonization of the Peyer's patches (PP), mesenteric lymph nodes (MLN), spleen, and liver 48 h after intragastric infection. n = 5. Top panels A-C: CFU/mg recovered for WT and chitinase-deficient *S*. Typhimurium. Bars represent geometric mean ± geometric SD. Bottom panels A-C: the same data expressed as a competitive index, CI = (WT colonization/mutant colonization)/ (WT inoculum/mutant inoculum). Bars represent geometric mean ± geometric SD. Statistics: (Top Row) Paired *t*-test was performed with the absolute values (Bottom Row) the competitive index was analyzed with a one-sample *t*-test against our null hypothesis (competitive index = 1). * = p<0.05.

glycan residues, as there are only minor differences in the abundances of individual glycans during infection with the single-deletion strains (Fig 6C and S4 Table).

We also found multiple glycans that increased in abundance upon infection with WT *S*. Typhimurium but not during infection with the chitinase-deficient strains (Fig 6D and 6E). Many of these are high molecular-weight complex or hybrid glycans (53100, 46110, 56040, 56100, 36100, 58100, 56120) (Fig 6D), including many glycans that contain Lewis X/A structures (55110, 57100, 66100, 67100, 67120) (Fig 6E and 6F). The upregulation of these glycans during WT *S*. Typhimurium infection indicates modulation of the expression of glycans by host cells. Since these changes do not occur during infection with chitinase-deficient strains, the detection of *S*. Typhimurium chitinases or their enzymatic activity by host cells may drive these glycomic changes. Overall, this data suggests that *S*. Typhimurium chitinases modulate the surface glycome during infection via direct enzymatic activity or indirect induction of host cell glycan expression to enhance the adhesion to and invasion of intestinal epithelial cells (Fig 7).

## Discussion

Bacterial chitinases have recently been recognized as virulence factors for various pathogenic species [39]. However, studies of *S*. Typhimurium chitinases have been mostly limited to the

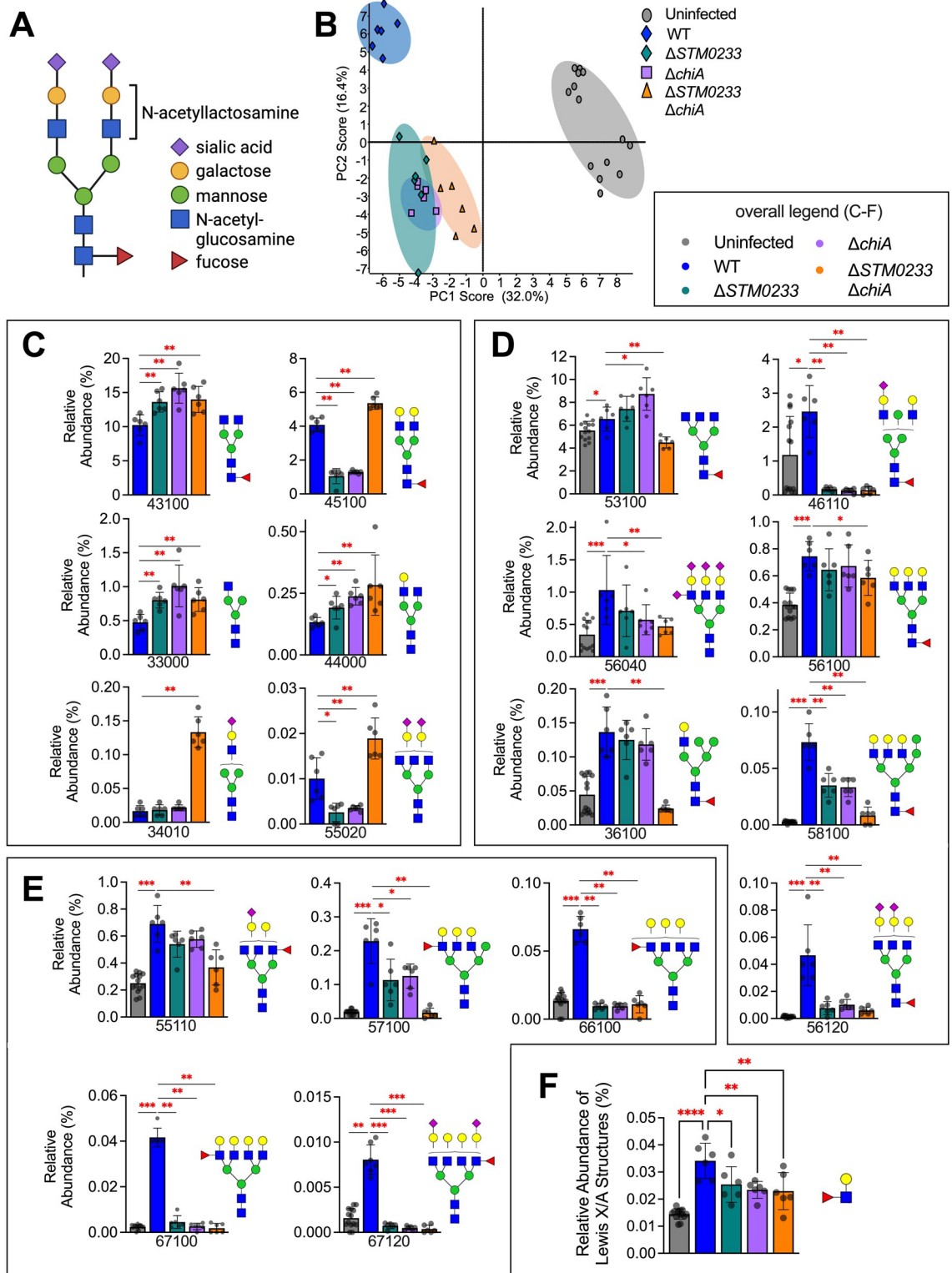

**Fig 6. *S*. Typhimurium chitinases induce specific changes in the intestinal glycome during infection.** (A) Common *N*-linked glycan structure. (B) Principal component analysis of the surface glycome composition of uninfected IPEC-1 cells and cells infected with WT *S*. Typhimurium or the chitinase-deficient strains. Uninfected n = 14, WT n = 6, Δ*STM0233* n = 6, Δ*chiA* n = 6, Δ*STM0233* Δ*chiA* n = 6. (C) Relative abundance of glycan species that increase during infection with Δ*STM0233* Δ*chiA* compared to WT *S*. Typhimurium. (D) Relative abundance of selected glycans species that increase during WT infection compared to uninfected but do

not increase during Δ*STM0233* Δ*chiA* infection (E) Relative abundance of selected Lewis X/A-containing glycans species that increase during WT infection compared to uninfected but do not increase during Δ*STM0233* Δ*chiA* infection. (F) Overall relative abundance of Lewis X/A structures on all glycans. A five-digit code is used to represent *N*-glycan structures. The 1st number denotes the number of hexosamine (HexNAc), i.e. N-acetyl glucosamine. The 2nd number denotes the number of hexose (Hex), i.e. mannose and galactose. The 3rd number denotes the number of fucose (DeoxyHex). The 4th and 5th numbers represent the number two different sialic acid structures, N-acetylneuraminic acid (NeuAc) and N-glycolylneuraminic acid (NeuGc), respectively. Bars represent mean ± SD. Statistics: (C-D) Mann-Whitney *U* test (F) One-way ANOVA with Dunnett's multiple comparison test. * = p<0.05, ** = p<0.01, *** = p<0.001.

enzymatic activity of ChiA, which was found to cleave chitin, LacNAc, and LacdiNAc molecules [22,23]. A functional role for *S.* Typhimurium chitinases during infection has been suggested based on the observations that *chiA* is upregulated during the infection of epithelial cells, murine macrophages, and the chicken gastrointestinal system [19–21]. One study, therefore, investigated ChiA's role in host cell invasion and pathogenicity [40]. Deletion of ChiA resulted in only slightly reduced invasion of non-intestinal epithelial cells and no competitive advantage of wild type *S.* Typhimurium over ChiA-deficient *S.* Typhimurium in a mixed infection mouse model, questioning the relevance of *S.* Typhimurium chitinases for pathogenicity. Our study corroborates these results, as we also show that the role of chitinases is cell-type specific and not apparent in mixed infections. Since our data indicate a functional redundancy of ChiA and STM0233, the deletion of ChiA alone may not have been sufficient to observe a phenotype with this study's models. Therefore, by using different experimental conditions and examining strains deficient in both chitinases, we were able to elucidate a role for *S.* Typhimurium chitinases during infection.

One role that has been identified for chitinases in other pathogens is the facilitation of binding to intestinal epithelial cells. ChiA of AIEC is known to enhance the adhesion to intestinal epithelial cells [16]. Chitin-binding proteins of *Serratia marcescens* (Cbp21) and *Vibrio cholera* (GbpA) have also been shown to promote adherence to intestinal epithelial cells [17,41]. Here, we demonstrate that both *S.* Typhimurium chitinases are involved in adhesion to intestinal epithelial cells (Fig 1D and 1E). This role in adhesion likely explains the invasion defect of the chitinase-deficient strains (Figs 1B and 1C, and 2C), as adhesion is a prerequisite of *S.* Typhimurium invasion [30]. Interestingly, *S.* Typhimurium chitinases may play a greater role in the adhesion to small intestinal tissue than colonic tissue, based on the larger adhesion/invasion defect *in vitro* and the lack of a colonic invasion defect *in vivo* (Figs 1B–1E and 2D). Previous literature has indicated that the binding of *S.* Typhimurium to a variety of epithelial cell lines is mediated by fimbriae specific to each cell type [42]. This suggests a potential role for chitinases as mediators for the binding of fimbriae specific to the small intestines, such as Pef [43]. Prior to this study, STM0233 had not been studied experimentally. Our data suggest that STM0233 may play a more significant role in adhesion and invasion than ChiA (Fig 1B–1E).

Based on the function of chitinases from other bacterial pathogens, there are a variety of possible explanations as to why *S.* Typhimurium chitinases enhance adhesion. For one, chitinases could be degrading mucins to provide access to the epithelial layer. *Legionella pneumophila* produces a chitinase (also named ChiA), which is not homologous to *S.* Typhimurium ChiA, that degrades mucins present in the lungs to enhance colonization [10,11]. It is now known that the binding of GbpA of *Vibrio cholerae* to intestinal mucins is responsible for enhanced adhesion to epithelial cells [15]. While interactions with intestinal mucins could potentially contribute to the *in vivo* colonization and invasion defect (Fig 2C), it does not explain our *in vitro* results (Fig 1C and 1E) as the IPEC-1 cell line is not known to be a mucin-producing cell line [44]. Therefore, we focused our study on the *N*-linked glycans present on IPEC-1 cells and did not explore possible interactions with the *O*-linked glycosylation of mucins.

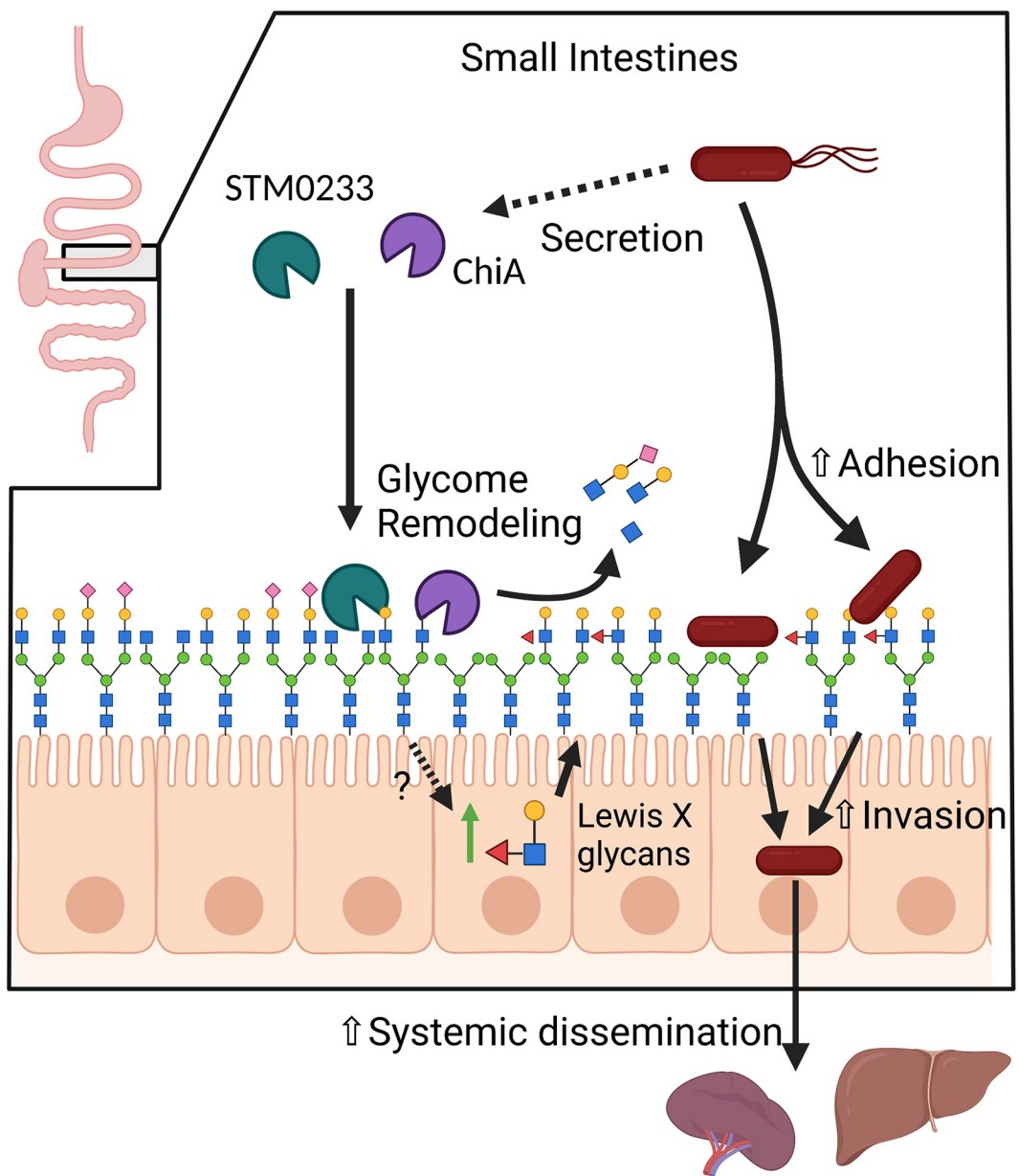

**Fig 7. Proposed Model.** STM0233 and ChiA are likely secreted into the small intestinal lumen during infection. STM0233 and ChiA remodel the surface glycome through the removal of GlcNAc residues of *N*-linked glycans. STM0233 and ChiA also stimulate the upregulation of Lewis X/A-containing glycans by host cells through an unknown mechanism. STM0233 and ChiA enhance *S*. Typhimurium adhesion to epithelial cells, likely due to the exposure of mannose and increase in Lewis X binding residues. Increased adhesion leads to increased invasion of intestinal epithelial cells and increased dissemination to the spleen and liver.

One possible role for chitinases is the liberation of nutritional resources from intestinal glycans to enhance luminal colonization. *Vibrio cholerae* produces a chitinase (ChiA2) that can degrade mucins, releasing saccharides that can be used as a carbon source [14]. Members of the commensal microbiota can also use intestinal glycans as a source of nutrients by producing various types of glycosyl hydrolases [45,46]. *S*. Typhimurium could be using glycans as a nutrient source. However, this seems unlikely since the invasion assays were performed in rich

media (Fig 1B–1D), where *S.* Typhimurium would not require an alternative carbon source to replicate efficiently.

Some bacterial chitinases are known to interact with the host immune system to promote infection. For example, one chitinase produced by *Listeria monocytogenes* (ChiA) was found to down-regulate *nos2* expression during murine infection [12]. *L. monocytogenes* strains lacking ChiA showed decreased colonization of the spleen and liver during murine infection [12,13]. This observation indicated an extraintestinal role for *L. monocytogenes* chitinases in modulating inflammation. In our mouse model of *S.* Typhimurium infection, we detected a minor increase in *Il17a* expression in chitinase-deficient strain infected mice but no changes in the expression of *nos2* or other innate immune genes (Fig 4A–4L). Even though chitinase-deficient strains displayed a colonization defect for the spleen and liver with an intestinal infection model (Fig 3A), there was no colonization defect when *S.* Typhimurium was delivered directly to the peritoneal cavity (Fig 3B). With this consideration, *S.* Typhimurium chitinases are likely specifically required for small intestinal infection through a mechanism that does not involve innate immune modulation.

Another possible mechanism for enhanced adhesion is through interactions between bacterial chitinases and the host protein Chitinase 3 like 1 (CH3L1). CH3L1 is a chitin-binding protein expressed by various cell types, such as intestinal epithelial cells [18] and macrophages [47], and is associated with inflammatory bowel disease [48,49]. Both CBP21 of *Serratia marcescens* and ChiA of AIEC exploit intestinal CH3L1 expression to adhere to epithelial cells [16,17]. CH3L1 appears to play a role in *Salmonella* infection, as the expression of CH3L1 by colonic epithelial cells was found to enhance *Salmonella* adhesion and invasion [18]. Our study has not explored the possibility that *Salmonella* chitinases promote adhesion by interacting with CH3L1 in a similar manner as CBP21 and ChiA of AIEC.

A likely target for bacterial chitinases is the N-linked surface glycoproteins expressed by host cells. *S.* Typhimurium is already known to trigger the removal of sialic acid during infection of colonic cells, which is likely to be mediated by sialidases expressed by *S.* Typhimurium [26,50]. Several other glycosyl hydrolases expressed by *S.* Typhimurium have also been implicated in the modulation of specific glycan species [25]. A previous study showed that WT and the invasion deficient Δ*invA* strain induce similar changes in the surface glycome of Caco-2 cells [26]. This similarity suggests that *S.* Typhimurium invasion does not influence the abundance of glycans on intestinal epithelial cells. We used a different intestinal epithelial cell line (IPEC-1), and it is possible that the response to *S.* Typhimurium invasion differs and that these cells modulate their glycome dependent on invasion rates, but we did not explore this in this study. Even though ChiA has demonstrated activity towards LacNAc residues [22,23], we did not detect broad removal of LacNAc residues during infection (S4 Table). Instead, we saw the abundance of specific GlcNAc-containing glycans increase during infection with the chitinase-deficient strains (Fig 6C). This pattern indicates that chitinases expressed by WT *S.* Typhimurium remove these residues, while other GlcNAc-containing glycans are unaffected. Chitinases may only be required for the removal of GlcNAc residues on specific glycoproteins that directly facilitate invasion. This hypothesis is supported by previous observations that the type 1 fimbria, FimH, promotes *S.* Typhimurium adhesion and invasion of M cells by specifically binding mannose-containing *N*-linked glycans of surface Glycoprotein 2 (GP2) [51,52]. It has also been shown that the binding of AIEC ChiA to CH3L1 is specifically dependent on the *N*-glycosylation of CH3L1 [16]. So chitinases may not be required to have broad activity towards all GlcNAc residues. Instead, they could specifically target glycan species that would markedly contribute to invasion.

Furthermore, our data also indicates that host cells are potentially detecting the activity of *S.* Typhimurium chitinases and are modulating their glycome to compensate. We observed an

increase in high molecular-weight complex glycans during infection with WT *S.* Typhimurium, potentially driven by host cell regulation (Fig 6C). Specifically, there was an increase in additional fucose subunits on terminal LacNAc residues forming Lewis X/A structures (Fig 6D). Increased fucosylation has been observed in a previous glycomic analysis of *S.* Typhimurium infection, which also identified increased expression of the host cell fucosyltransferases that would be responsible for this fucosylation [25]. It seems that this upregulation of fucosylation is dependent on *S.* Typhimurium chitinase activity, as the relative abundance of Lewis X/A-containing glycans during chitinase-deficient strain infection is comparable to uninfected cells (Fig 6E and 6F). It is important to note that *S.* Typhimurium expresses a fimbrial adhesin, Pef, that binds Lewis X structures and is specifically required for small intestinal adhesion [43,53]. Our glycomics analysis does not allow differentiation between the β(1,3)-fucosylation of Lewis X glycans and the β(1,4)-fucosylation of Lewis A glycans. If Lewis X glycans are being upregulated, then *S.* Typhimurium may exploit these Lewis X structures as an alternative binding site, which would explain the tissue tropism of chitinase-mediated invasion of the small intestines (Fig 2C).

Here we elucidated novel roles for *S.* Typhimurium chitinases (ChiA and STM0233) in promoting infection by enhancing small intestinal invasion. We have demonstrated that chitinases promote the adhesion to epithelial cells, which likely drives the enhanced invasion. We also showed that chitinases are required for optimal early small intestinal infection and promote increased dissemination of *S.* Typhimurium into systemic circulation. Increased invasion is linked to modulations of the epithelial cell glycome induced by *S.* Typhimurium chitinases. Chitinases cause alterations in the abundance of specific GlcNAc-containing glycans that would indicate chitinase-mediated cleavage and increase the abundance of Lewis X/A-containing glycans, likely by stimulating host cell expression.

## Materials and methods

### Ethics statement

Mouse experiments were performed in accordance with protocols and guidelines approved by the Institutional Animal Care Committee (20–016) of the University of Illinois Chicago.

### Bacterial strains

All strains were grown while shaking at 200 rpm in 5 mL of Luria-Bertani (LB) broth (BD Diagnostic Systems) for 16 h at 37°C unless otherwise stated (S1 Table).

### Mutant generation

Wild-type *S.* Typhimurium IR715 was used to generate all deletion strains. Deletion constructs containing an antibiotic resistance cassette flanked by sequences homologous to 1kb upstream and downstream of the target gene on a pGP704 suicide vector backbone were generated by Gibson assembly (S1 Table). Q5 High-Fidelity DNA Polymerase (New England Biolabs) was used for all cloning PCR reactions. The genome of IR715 was used to generate the homologous flanking regions for each gene. The primers used to generate the flanking regions of *chiA* were chiA_FR1_fwd, chiA_FR1_rev, chiA_FR2_fwd#2, and chiA_FR2_rev#2. The flanking region for *STM0233* was generated with the primers STM0233_FR1_fwd, STM0233_FR1_rev, STM0233_FR2_fwd, and STM0233_FR2_rev. The flanking region for *invA* was generated with the primers invA-LB_FW, invA-LB_RV, invA-RB_FW, and invA-RB_RV. The *STM0233* and *invA* deletion constructs contained a chloramphenicol resistance cassette, while the *chiA* deletion construct contained a kanamycin resistance cassette. The chloramphenicol resistance cassette was generated by PCR using pKD3 as a template and primers CmR_fwd and CmR_rev

for the *STM0223* deletion construct and invA-CmR_FW and invA-CmR_RV for the *invA* deletion construct. The kanamycin resistance cassette was generated by PCR using pKD4 as the template and the primers chiA_KanR_fwd and chiA_KanR_rev#2. pGP704 was digested with the restriction enzymes SalI and EcoRV. The two flanking regions, the antibiotic resistance cassette, and the vector backbone were ligated together using the Gibson Assembly Master Mix (NEBuilder), generating the plasmids pGP704-*STM0233*::Cm, pGP704-*chiA*::Kan, and pGP704-*invA*::Cm. *Escherichia coli* S17 λpir cells were transformed with either pGP704-*STM0233*::Cm, pGP704-*chiA*::Kan, or pGP704-*invA*::Cm and selected for on LB agar with carbenicillin (0.1 mg/mL). The transformation was done by electroporation (Gene Pulser II, Biorad) using 1 μL of a 1:3 dilution of the assembly reaction (in ultrapure H$_2$O). Single colonies of the resulting *E. coli* S17 λpir cells containing either pGP704-*STM0233*::Cm, pGP704-*chiA*::Kan, or pGP704-*invA*::Cm were grown in LB broth with carbenicillin (0.1 mg/mL). These *E. coli* cultures were plated on LB agar with a culture of IR715 at a 1:1 ratio for conjugation. Plates were incubated at 37˚C for 16 h, allowing deletion constructs to be transferred to IR715 and the target genes to be replaced with the antibiotic resistance cassette by homologous recombination. The bacterial lawns were harvested, and the resulting deletion strains, *ΔSTM0233* and *ΔinvA*, were selected for on LB agar with nalidixic acid (0.05 mg/mL) and chloramphenicol (0.03 mg/mL). The *ΔchiA* strain was selected for on LB agar with nalidixic acid (0.05 mg/mL) and kanamycin (0.1 mg/mL). Two PCR reactions were used to confirm that the antibiotic resistance cassettes were inserted into the correct location. The primer sets used for these reactions were C2 and STM0233_FR1_out_Fw, and C1 and STM0233_FR2_out_Rv for Δ*STM0233*. The primer sets used for Δ*chiA* were chiA_out_Fw and K1_(Kan_Rv), and chiA_out_Rv and K3_(Kan_Fw). The primer sets used for Δ*invA* were invA_LB_chk_FW and C3, and C1 and invA_RB_chk_RV. The chitinase deletion strains were further examined to confirm that the target genes were deleted. The presence of *STM0233* was detected with the primer pair STM0233_pres_Fw and STM0233_pres_Rv, and *chiA* was detected with the primer pair chiA_pres_Fw and chiA_pres_Rv (S2 Table).

The *S.* Typhimurium IR715 Δ*STM0233* Δ*chiA* strain was generated by transduction using the P22 *HT105/1 int-201* bacteriophage (S1 Table). A culture of the IR715 Δ*chiA* strain was infected with P22 *HT105/1 int-201* for 8 h at 37˚C. The phage was isolated by the centrifugation of the infected culture, and chloroform was added to the collected supernatant. Bacteriophage transduction was performed by plating 200 μL of a 16 h IR715 Δ*STM0233* culture and 1–20 μL of the isolated P22 phage onto LB agar with kanamycin (0.1 mg/mL). Plates were incubated at 37˚C overnight, and the resulting colonies of transductants were cross-streaked against the P22 H5 phage on Evans blue-Uranine (EBU) agar to isolate phage-free true lysogens.

Complementation of the chitinase deletion strains was done by transposon-mediated insertion of the full chitinase gene and endogenous promoter into the Tn7 locus [54]. Chitinase genes and their endogenous promoters were amplified using Q5 High-Fidelity DNA Polymerase (New England Biolabs) with wild-type *S.* Typhimurium IR715 as the template and the primer sets STM0233_Tn7Com_FW and STM0233_Tn7Com_RV or chiA_Tn7Com_FW and chiA_Tn7Com_RV, which carry restriction sites for either XmaI or XhoI on their 5' end. The gene fragments were sub-cloned into Zero Blunt TOPO PCR vector (Invitrogen) and electroporated into One Shot Top10 cells (Invitrogen). The plasmid containing the cloned chitinase gene was isolated using the Qiaprep spin miniprep kit (Qiagen). Restriction digestion of the plasmid was done with XhoI and XmaI, and the chitinase gene fragment was isolated via gel extraction with the QIAQuick Gel Extraction kit (Qiagen). pGRG36 (a gift from Nancy Craig [54]) was also digested with XhoI and XmaI and extracted via gel extraction using the QIAEX II gel extraction kit (Qiagen). The digested pGRG36 and cloned chitinase gene were ligated

together using ElectroLigase (New England Biolabs) and used to transform One Shot Electro-competent cells. Transformants were selected for by growing at 32˚C on LB agar with carbenicillin (0.1 mg/mL). pGRG36-*STM0233* and pGRG36-*chiA* were isolated using the Plasmid Midi kit (Qiagen), and Sanger sequencing was performed to confirm that the cloned chitinase genes had the correct sequence. Sanger sequencing was performed by the Genome Research core at the University of Illinois Chicago. The chitinase deletion strains (BL114, BL122, and BL130) were transformed with pGRG36-*STM0233* or pGRG36-*chiA*. Transformants were isolated and grown in an LB culture at 32˚C overnight. Overnight cultures were serially diluted in PBS and plated for single colonies on LB agar plates. Plates were incubated at 42˚C overnight to block plasmid replication (pGRG36 has a temperature-sensitive origin of replication). Single colonies were streaked on LB agar and incubated at 42˚C overnight to ensure plasmid loss. Insertion of the chitinase genes in the Tn7 locus was confirmed by colony PCR. The presence of *STM0233* was detected with the primer pair STM0233_pres_Fw and STM0233_pres_Rv, and *chiA* was detected with the primer pair chiA_pres_Fw and chiA_pres_Rv. A primer set that flanks the Tn7 insertion site (Tn7_ck_FW and Tn7_ck_RV) was used to confirm the insertion site.

## Tissue culture

The T84 colonic epithelial cell line (ATCC Cat# CCL-248; RRID:CVCL_0555) and IPEC-1 small intestinal epithelial cell line (DSMZ Cat# ACC-705, RRID:CVCL_2245) were grown in T75 flasks using 20 mL DMEM/F12 + 10% FBS + 1x antibiotic/antimycotic (Gibco). Cultures were incubated at 37˚C with 5% $CO_2$. Media was replaced every other day. Cell cultures were split with 0.25% trypsin + EDTA (Gibco) when cells reached 80% confluency.

## Invasion assay

T84 or IPEC-1 cells were seeded onto a 24-well plate at a density of $5x10^5$ cells/well with media lacking antibiotics/antimycotics and incubated overnight at 37˚C. *S.* Typhimurium strains were grown for 16 h in liquid LB without shaking at 37˚C. Bacterial cell number was quantified by measuring the $OD_{600}$ of the cultures. $1x10^9$ cells of each strain were centrifuged, resuspended in DMEM/F12, and serially diluted. Epithelial cells were infected with $5x10^5$ cells of *S.* Typhimurium (multiplicity of infection (MOI) = 1). Infected cells were incubated at 37˚C for 1 h. The inoculum was serially diluted and plated on LB agar to confirm bacterial numbers. After infection, the media was removed via vacuum, and wells were washed 3 times with 500 μL phosphate-buffered saline (PBS). 500 μL of DMEM/F12 + 10% FBS + 0.1 mg/mL gentamicin was added to the wells and incubated at 37˚C for 1 h to kill extracellular bacteria. After incubation, the wells were washed with PBS and lysed by incubation with 1% Triton X-100 for 5 mins. Cells were disrupted and harvested by scraping wells and pipetting, were serially diluted, and plated on LB agar to quantify bacterial cells that invaded. The percentage of cells recovered relative to the inoculum was calculated.

## Adhesion assay

This assay was adapted from a previous study [55]. T84 cells were seeded onto a 24-well plate at a density of $2x10^6$ cells/well to achieve confluency and prevent nonspecific binding of *S.* Typhimurium to the bottom of the well. IPEC-1 cells were seeded at a density of $5x10^5$ cells/well, which was sufficient to achieve confluency. The media used lacked antibiotics/antimycotics, and the cells were incubated overnight at 37˚C. *S.* Typhimurium strains were grown in liquid LB without shaking for 16 h at 37˚C. T84 and IPEC-1 cells were incubated in DMEM/F12 + 10% FBS + 2 μg/ml Cytochalasin D (Sigma-Aldrich) at 37˚C for 1 h to block actin-dependent

invasion. The $OD_{600}$ was measured for each *S.* Typhimurium culture, and $1x10^9$ cells of each strain were centrifuged, resuspended in DMEM/F12, and serially diluted. While in the presence of Cytochalasin D, epithelial cells were infected at a MOI = 1. Infection was carried out for 30 minutes, and epithelial cells were washed 4 times with PBS and lysed with 1 mL 1% Triton X-100. Cells were disrupted and harvested by scraping wells and pipetting, serially diluted, and plated on LB agar to quantify adherent bacterial cells. The percentage of cells recovered relative to the inoculum was calculated.

### *In vitro* growth curve

Colloidal chitin was made based on a previous study [56]. Crab shell flakes were ground by mortar and pestle and sieved through a 130 mm two-piece polypropylene Büchner filter. 20 g of sieved crab shell flakes were placed in a beaker, and 150 mL of 12 M HCl was added slowly with continuous stirring. The chitin-HCl mixture was stirred every 5 min over the course of an hour. The mixture was then passed through 8 layers of cheesecloth to remove large chunks into a 2 L plastic beaker. 2 L of ice-cold MilliQ water was added and incubated at 4˚C for 16 h. After incubation, 3 L of tap water was passed through the colloidal chitin cake on two layers of coffee filter paper in a Büchner funnel connected to a vacuum filtration flask until the pH of the filtrate was 7.0. Excess moisture was removed by pressing the colloidal chitin cake between coffee filter paper. The colloidal chitin cake was sterilized in an autoclave and used to make M9 + 0.4% colloidal chitin medium (M9 + chitin). The bacterial concentrations of 16 h *S.* Typhimurium cultures grown in LB were determined by measuring $OD_{600}$. 20 mL of LB or M9 + chitin was inoculated with $1 \times 10^5$ cells of *S.* Typhimurium. Cultures were incubated shaking at 37˚C, and samples were taken at indicated time points, serially diluted in PBS, and plated on LB agar.

### Mouse infection

A streptomycin-pretreatment mouse model was used for the *in vivo* infections [33], where each mouse is considered an experimental unit. Single infection experiments were repeated twice with 3–5 mice per treatment group. 8 week-old female C57BL/6 mice were purchased from a maximum barrier facility at Jackson Laboratory and were free of *Enterobacteriaceae* (tested by plating fecal samples on MacConkey agar). Mice were housed for 1 week after arrival to allow for their acclimation and given water and food *ad libitum* (mice were fed Teklad Irradiated LM-485 mouse diet 7912). Mice were treated with 100 μL of 200 mg/mL streptomycin (Calbiochem) in water by oral gavage. 24 h after treatment, mice were infected by oral gavage with 100 μL of $1x10^{10}$ cells/mL of *S.* Typhimurium in LB broth from a 16 h culture. For co-infections, mice were infected with a 1:1 ratio of both *S.* Typhimurium strains. The *S.* Typhimurium strains used carry plasmid pHP45Ω, which confers resistance to streptomycin and carbenicillin. Fecal samples were collected at 8 hpi and 24 hpi and either snap-frozen or plated on LB agar + carbenicillin (0.1 mg/mL). At 48 hpi (or 96 hpi), mice were euthanized by $CO_2$ asphyxiation and subsequent cervical dislocation. Immediately after euthanasia, the Peyer's patches, mesenteric lymph nodes, spleen, liver, and luminal samples from the colon and ileum were collected, homogenized, and plated on LB agar with the appropriate antibiotic. Colony-forming units were normalized to the weight of each sample (mg). Cecum, ileum, and luminal colon samples were snap-frozen. Ileum and cecum samples were collected and fixed in formalin for histological analysis. Tissue was collected from the terminal ileum and proximal colon, and a gentamicin protection assay was performed. The tissue was incubated in PBS + 0.1 mg/mL gentamicin for 30 mins, washed with PBS, homogenized, and plated on LB agar + carbenicillin (0.1 mg/mL). For co-infection, the mutant strain was quantified by plating on LB agar

+ chloramphenicol (0.03 mg/mL). WT colonization was calculated by subtracting the mutant CFUs from the CFUs appearing on LB agar + carbenicillin. A competitive index was calculated by dividing CFU/mg of recovered WT by CFU/mg of recovered mutant. This competitive index was corrected based on the ratio of each strain in the inoculum, determined by serial dilution followed by plating on LB agar. Corrected CI = (WT colonization/mutant colonization)/ (WT inoculum/mutant inoculum).

For intraperitoneal infection, 9 week-old female C57BL/6 mice were infected via intraperitoneal injection with 100 μL of $1x10^5$ cells/ml of *S*. Typhimurium in PBS from a 16 h culture. Mice were sacrificed at 24 hpi or 48 hpi, and the spleen and liver were collected. Samples were homogenized and plated on LB agar and the appropriate antibiotic to quantify colonization.

## Histopathology

Tissue sections were fixed with formalin and embedded in paraffin wax. Embedding was done by the Research Histology core at the University of Illinois Chicago. The tissue was then sectioned via microtome and transferred to slides. Before staining, deparaffinization was performed. The tissue slides were immersed in xylene for 10 min, 100% ethanol for 10 min, 90% ethanol for 2 min, 70% ethanol for 2 min, and PBS for 5 min. The tissue slides were then stained with hematoxylin for 30 seconds, washed with tap water, then stained with eosin for 10 min. Slides were dehydrated by immersion in serial increases of ethanol concentrations (50%-100%), then immersed in xylene. Coverslips were then mounted to the tissue slides and allowed to dry.

Tissue sections were scored for pathology by a board-certified pathologist in a blinded fashion following an approach established by Barthel and colleagues [33] as summarized below.

Submucosal edema was scored as follows: 0 = no pathological changes; 1 = mild edema (submucosa accounts for <50% of the diameter of the entire intestinal wall [tunica muscularis to epithelium]); 2 = moderate edema; the submucosa accounts for 50 to 80% of the diameter of the entire intestinal wall; and 3 = profound edema (the submucosa accounts for >80% of the diameter of the entire intestinal wall).

Polymorphonuclear granulocytes (PMN) in the lamina propria were enumerated in 10 high-power fields (x400 magnification), and the average number of PMN/high-power fields was calculated. The scores were defined as follows: 0 = <5 PMN/high-power field; 1 = 5 to 20 PMN/high-power field; 2 = 21 to 60/high-power field; 3 = 61 to 100/high-power field; and 4 = >100/high-power field. Transmigration of PMN into the intestinal lumen was consistently observed when the number of PMN was >60 PMN/high-power field.

The average number of goblet cells per high-power field (magnification, x400) was determined from 10 different regions of the cecal epithelium. Scoring was as follows: 0 = >28 goblet cells/high-power field (magnification, x400); 1 = 11 to 28 goblet cells/high-power field; 2 = 1 to 10 goblet cells/high-power field; and 3 = <1 goblet cell/high-power field.

Epithelial integrity was scored as follows: 0 = no pathological changes detectable in 10 high-power fields (x400 magnification); 1 = epithelial desquamation; 2 = erosion of the epithelial surface (gaps of 1 to 10 epithelial cells/lesion); and 3 = epithelial ulceration (gaps of >10 epithelial cells/lesion).

Two independent scores for submucosal edema, PMN infiltration, goblet cells, and epithelial integrity were averaged for each tissue sample. The combined pathological score for each tissue sample was determined as the sum of these averaged scores. It ranges between 0 and 13 arbitrary units and covers the following levels of inflammation: 0 intestine intact without any signs of inflammation; 1 to 2 minimal signs of inflammation; 3 to 4 slight inflammation; 5 to 8 moderate inflammation; and 9 to 13 profound inflammation.

## RNA extraction

*In vivo*: Cecum and ileum samples collected from mice 48 hpi were homogenized by mortar and pestle and liquid nitrogen. Because mice were treated with streptomycin 24 h prior to infection, we collected cecum and ileum samples from uninfected mice 72 h after streptomycin treatment as a control. The homogenate was transferred to 1 mL of Tri-Reagent (Molecular Research Center) for RNA extraction. RNA was extracted with 0.1 mL of bromo-3-chloropropane, centrifuged, and the upper phase was precipitated with 0.5 mL isopropanol. After centrifugation, pellets were washed twice with 1 mL of 75% ethanol in RNase-free water. The RNA pellet was then resuspended in RNase-free water. RNA was treated with DNase using the Turbo DNA-free kit (Invitrogen). For fecal samples, RNA was extracted from snap-frozen luminal colon samples collected during the single infection mouse experiments. RNA extraction was performed using the Qiagen RNeasy PowerMicrobiome kit, and DNase treatment was performed with the Turbo DNA-free kit.

*In vitro*: A 16 h culture of WT *S*. Typhimurium was sub-cultured (1:100) into LB or DMEM/12 +10% FBS. Cultures were incubated for 3 h at 37˚C, shaking at 200 rpm. Culture cell concentration was measured by $OD_{600}$, and $1x10^9$ cells were pelleted by centrifugation. RNA extraction was performed on the pellet using the Invitrogen RiboPure Bacteria kit. RNA extraction was followed by DNase treatment using the Turbo DNA-free kit.

## RT-qPCR

Reverse transcription was performed with the High Capacity cDNA Reverse Transcription Kit (Applied Biosystems). For *S*. Typhimurium RNA, reactions were also performed without the addition of reverse transcriptase to confirm that there was no amplification of DNA in qPCR reactions. 1000 ng of RNA was used for the reverse transcription reaction. The reverse transcription cycle consisted of 10 minutes at 25˚C followed by 120 minutes at 37˚C and 5 minutes at 85˚C. qPCR was performed using the Fast SYBR Green Master Mix (Applied Biosystems) on the Viia7 Real-time PCR system at the Genome Research core at the University of Illinois at Chicago. The qPCR reaction cycle consisted of 20 seconds at 95˚C followed by 40 cycles of 3 seconds at 95˚C and 30 seconds at 60˚C. Reactions were performed in duplicate. Relative expression was calculated based on the ΔCT values. For the analysis of chitinase expression, the CT value of the house-keeping gene (*gmk*), which has been shown to be stably expressed across a variety of conditions [57], was subtracted from the CT value of the gene of interest, giving the ΔCT value. Relative expression = $2^{(-\Delta CT)}$. *In vivo* chitinase gene expression was compared to expression in LB broth by subtracting the ΔCT value of LB broth from the ΔCT of fecal samples. Relative expression = $2^{(-\Delta\Delta CT)}$. For analysis of murine immune gene expression, the CT value of the housekeeping gene (*actb*) was subtracted from the CT value of the gene of interest, giving the ΔCT value. The ΔCT value of uninfected mice was subtracted from the ΔCT value of infected mice, giving the ΔΔCT value. Relative expression = $2^{(-\Delta\Delta CT)}$ (S3 Table).

## Glycome analysis

Infection of IPEC-1 cells was carried out similar to the invasion assays at an MOI of 1:1000. Two biological replicates with three technical replicates each were used for each infected sample. Biological replicates consisted of culturing and seeding IPEC-1 cells independently of each other. Uninfected samples also consisted of two independently cultured biological replicates, with 7 technical replicates for each. After infection, wells were washed twice with PBS, and cells were frozen at -80˚C. Cells were thawed and washed twice with PBS to remove the lysed cell debris and cytoplasmic proteins. Next, 200 μL PBS and 3 μL PNGase F were added to each well and incubated at 37˚C for 18 hrs to release surface *N*-glycans. A sealing film (Axygen

PCRSPS) was employed to cover the well plate to prevent evaporation. The released *N*-glycans solution was collected, and the well was washed with 200 μL PBS. The wash solution was collected and combined with the previously released *N*-glycan solution, dried, and redissolved in 100 μL water. *N*-glycans were then dialyzed against a 500–1000 MWCO dialysis membrane to remove salts and small molecules. The dialyzed sample was then reduced and permethylated prior to LC-MS/MS analysis, as previously reported [58–60]. Briefly, the dried sample was dissolved in 10 μL borane-ammonia complex solution (10 mg/mL) and incubated in a 60˚C water bath for 1h. After reduction, 1 mL of methanol was added to each sample and dried. The methanol addition-dry cycle was repeated 3 times to remove borates. Next, a spin column was packed with sodium hydroxyl beads (suspended in DMSO) and washed twice with 200 μL DMSO. Reduced glycans were resuspended in 30 μL DMSO, 1.2 μL water, and 20 μL iodomethane. The sample was loaded on the column and incubated for 25 min. Then, 20 μL iodomethane was added to each column and incubated for 15 min. After incubation, permethylated glycan solution was collected by centrifuging at 1,800 rpm. The column was then washed with 30 μL acetonitrile (ACN), and the ACN solution was combined with permethylated glycan solution and dried. The reduced and permethylated sample was ready for LC-MS/MS analysis.

Samples were analyzed using an UltiMate 3000 nanoLC system coupled to an LTQ Orbitrap Velos mass spectrometer. The samples were resuspended in 8 μL solution (20% ACN, 80% water, 0.1% formic acid) and injected 6 μL. A PepMap trap column (75 μm* 2 cm, C18, 3 μm, Thermo) was used for online purification. The *N*-glycomic analysis was performed on a PepMap column (75 μm * 15 cm, C18, 2 μm, Thermo) at 55˚C at 0.35 μL/min flow rate. A gradient of mobile phase solvents A (98% water with 0.1% FA) and B (98% ACN with 0.1% FA) was used as follows: 0–10 min, 20% B; 10–11 min, 20% - 42% B; 11–48 min, 42% - 55% B; 48–49 min, 55% - 90% B; 49–54 min, 90% B; 54–55 min, 90% - 20% B; 55–60 min, 20% B. The MS was performed in positive mode. The full MS scan had a range of 700−2000 *m/z* at a mass resolution of 100,000. The CID (collision-induced dissociation) was used for MS$^2$ at a normalized collision energy of 35, activation Q of 0.25, and activation time of 10 ms. The data were first processed by MultiGlycan software [61], then manually checked via full MS and MS$^2$ to remove false positives. N-glycan biosynthesis is performed by a series of glycosidases and glycosyltransferases, and monosaccharides are sequentially added through the action of specific transferases [62]. Therefore, the assignment of putative *N*-glycan structures can be achieved through matching full MS to *N*-glycan compositions followed by the structural confirmation via MS$^2$, as the examples shown in (S8 Fig).

## Statistical analysis

Statistical analysis was performed with Graphpad Prism software v9 (RRID:SCR_002798). Data were tested for normality using the Shapiro-Wilk test and confirmed visually via QQ plot. For data found to follow a lognormal distribution, data were transformed to their natural log values before analysis. Unless otherwise specified, all multiple comparison analyses compared samples to the WT control. For invasion assays, the percent recovery was analyzed with a one-way ANOVA with Dunnett's multiple comparison test. For adhesion assays, the percent recovery was analyzed with a mixed-effect analysis with Dunnett's multiple comparison test. For mouse single infection experiments, a one-way ANOVA with Dunnett's multiple comparison test (Intragastric) or an unpaired *t*-test was performed (Intraperitoneal). For mouse co-infection experiments, a paired *t*-test was performed with the absolute values, and the competitive index was analyzed with a one-sample *t*-test against our null hypothesis that the competitive index equaled one. For *in vitro* chitinase gene expression analysis, an unpaired *t*-test was

performed. *In vivo* relative expression of chitinase expression was analyzed with a one-sample *t*-test against our null hypothesis (relative expression = 1). For murine gene expression, a one-way ANOVA with Dunnett's multiple comparison test was performed. Analysis of growth curves was done with a mixed-effect analysis and Tukey's multiple comparisons test. For the glycome analysis, the relative abundance for each glycan was calculated by dividing the individual glycan abundance by the total glycan abundance. A Mann-Whitney *U* test was then performed comparing samples to WT or uninfected. The relative abundance of Lewis X/A structures was calculated by dividing the total number of glycans that contain these structures by the total glycan abundance of each sample. A one-way ANOVA with Dunnett's multiple comparison test was then performed. For all statistical tests, significance was set at $\alpha = 0.05$.

## Supporting information

**S1 Fig. Deletion of *S.* Typhimurium chitinases does not affect growth characteristics.** (A) Growth of *S.* Typhimurium strains in LB broth at 37˚C. WT n = 5, chitinase-deficient strains n = 3. Points represent geometric mean ± geometric SD. (B) Growth of *S.* Typhimurium strains in M9 minimal medium +0.4% colloidal chitin at 37˚C. n = 3. Points represent geometric mean ± geometric SD. Statistics: Mixed-effect analysis and Tukey's multiple comparisons test.
(TIFF)

**S2 Fig. Complementation of chitinase-deficient strains restores invasion.** Chitinase genes were inserted into the Tn7 locus of chitinase-deficient *S.* Typhimurium. Gentamicin protection assay of *S.* Typhimurium infected (MOI:1) small intestinal epithelial cells (IPEC-1). (A) Invasion of Δ*chiA* and *chiA* complemented strain. n = 8. (B) Invasion of Δ*STM0233* and *STM0233* complemented strain. n = 8. (C) Invasion of Δ*STM0233* Δ*chiA* (Δ2) and Δ2 strain complemented with *chiA* or *STM0233*. n = 6. Percent recovery of each strain was normalized to WT recovery. Bars represent mean ± SEM. Statistics: One-way ANOVA with Dunnett's multiple comparison test.
(TIFF)

**S3 Fig. Cytochalasin D treatment completely blocks *S.* Typhimurium invasion.** (A) Gentamicin protection assay of *S.* Typhimurium infected (MOI:1) colonic epithelial cells (T84) after treatment of epithelial cells with cytochalasin D (1–2 μg/mL). (B) Gentamicin protection assay of *S.* Typhimurium infected (MOI:1) small intestinal epithelial cells (IPEC-1) after treatment of epithelial cells with cytochalasin D (2 μg/mL). n = 2. Bars represent mean ± SD.
(TIFF)

**S4 Fig. *S.* Typhimurium chitinases are not required for late-stage infection.** (A) Streptomycin pre-treatment mouse model of 96 h *S.* Typhimurium infection. (B) Luminal samples from the terminal ileum were collected at 96 hpi to determine *S.* Typhimurium colonization. Invasion was determined with a gentamicin protection assay performed on the terminal ileum. WT n = 3, Δ*STM0233* n = 3, Δ*chiA* n = 4. (C) *S.* Typhimurium colonies recovered from fecal samples collected at 24, 48, and 72 hpi. Fecal samples at 96 hpi were collected directly from the lumen of the colon. Invasion of colonic tissue was determined with a gentamicin protection assay. WT n = 10, Δ*STM0233* n = 8, Δ*chiA* n = 8. (D) Colonization of the Peyer's patches, mesenteric lymph nodes, spleen, and liver after 96 h of intragastric infection. WT n = 10, Δ*STM0233* n = 8, Δ*chiA* n = 8. Bars represent geometric mean ± geometric SD. Statistics: (B-D) Stars indicate significance compared to the WT control by one-way ANOVA with Dunnett's multiple comparison test. * = $p < 0.05$.
(TIFF)

**S5 Fig. Representative images of cecum and ileum histology. Uninfected:** (A) & (B) Cecum–intestine intact without any signs of inflammation [A: low power, hematoxylin eosin, original magnification 40x; B: intermediate power, hematoxylin-eosin, original magnification 200x]. (C) & (D) Ileum–intestine intact without any signs of inflammation [C: low power, hematoxylin eosin, original magnification 40x; D: intermediate power, hematoxylin eosin, original magnification 200x]. **WT:** (A) & (B) Cecum–moderate inflammation (moderate submucosal edema with lamina propria neutrophilic infiltration with cryptitis, crypt abscess and decrease in goblet cells) [A: low power, hematoxylin-eosin, original magnification 40x; B: high power, hematoxylin-eosin, original magnification 400x]. (C) & (D) Ileum–intestine intact without any signs of inflammation [C: low power, hematoxylin-eosin, original magnification 40x; D: intermediate power, hematoxylin-eosin, original magnification 200x]. **Δ*STM0233*:** (A) & (B) Cecum–moderate inflammation (moderate submucosal edema with lamina propria neutrophilic infiltration with cryptitis, crypt abscess and decrease in goblet cells) [A: low power, hematoxylin-eosin, original magnification 40x; B: intermediate power, hematoxylin-eosin, original magnification 200x]. (C) & (D) Ileum–intestine intact without any signs of inflammation [C: low power, hematoxylin-eosin, original magnification 40x; D: intermediate power, hematoxylin-eosin, original magnification 200x]. **Δ*chiA*:** (A) & (B) Cecum–moderate inflammation (profound submucosal edema with lamina propria neutrophilic infiltration with cryptitis, crypt abscess and decrease in goblet cells) [A: low power, hematoxylin-eosin, original magnification 40x; B: intermediate power, hematoxylin-eosin, original magnification 200x]. (C) & (D) Ileum–intestine intact without any signs of inflammation [C: low power, hematoxylin-eosin, original magnification 40x; D: intermediate power, hematoxylin-eosin, original magnification 200x]. **Δ*STM0233* Δ*chiA*:** (A) & (B) Cecum–moderate inflammation (moderate submucosal edema with lamina propria neutrophilic infiltration with cryptitis, crypt abscess and decrease in goblet cells) [A: low power, hematoxylin-eosin, original magnification 40x; B: high power, hematoxylin- eosin, original magnification 400x]. (C) & (D) Ileum–intestine intact without any signs of inflammation [C: low power, hematoxylin-eosin, original magnification 40x; D: intermediate power, hematoxylin-eosin, original magnification 200x].
(TIFF)

**S6 Fig. Innate immune gene expression in infected ileal tissue.** (A-E) mRNA expression of innate immune genes extracted from ileal tissue of *S.* Typhimurium infected mice 48 hpi. Expression is normalized to the housekeeping gene *actb* as well as gene expression in uninfected mice. n = 8 per group. Bars represent geometric mean ± geometric SD. There was no statistical significance ($\alpha$ = 0.05) comparing groups to the WT control by one-way ANOVA with Dunnett's multiple comparison test.
(TIFF)

**S7 Fig. Chitinase-deficient strains maintained invasion defects for glycome analysis.** Gentamicin protection assay of *S.* Typhimurium infected IPEC-1 cells (MOI:1000) done concurrently with infection for the glycome analysis. n = 2. Bars represent mean ± SD.
(TIFF)

**S8 Fig. *N*-linked glycan identification using MS data.** Examples of (A) complex and (B) high-mannose *N*-glycan identification through full MS and MS2. Insets are full MS spectra showing isotopic distribution envelops. The *N*-glycan compositions are identified through full MS within an *m/z* accuracy of 10 ppm. The putative *N*-glycan structures are further confirmed by matching the fragment ions detected in MS2 spectra.
(TIFF)

**S1 Table. Bacterial strains and plasmids used.**
(XLSX)

**S2 Table. Primers used for cloning.**
(XLSX)

**S3 Table. Primers used for RT-qPCR.**
(XLSX)

**S4 Table. Relative abundance of all glycan species identified in glycome analysis.** Average relative abundance of glycan species after infecting IPEC-1 cells with *S*. Typhimurium for 1 h (MOI:1000). Uninfected n = 14, WT n = 6, Δ*STM0233* n = 6, Δ*chiA* n = 6, Δ*STM0233* Δ*chiA* n = 6. Relative abundance is calculated by dividing glycan species abundance by total glycan abundance. Raw data has been uploaded to GlycoPOST database [63], accession number GPST000225.
(XLSX)

**S1 Data. Numerical data used for statistical analysis.** Excel sheet containing the numerical data for Figs 1A–1E, 2B–2D, 3A and 3B, 4A–4M, 5A–5C, 6F, S1A and S1B, S2A–S2C, S3A and S3B, S4B–S4D, S6A–S6E, and S7.
(XLSX)

## Acknowledgments

We thank Clayton Wollner and Kristen Lednovich for technical assistance in initial experiments. We thank Dara Kiani, Amisha Rana, Kelly Perfecto, Kanchan Jaswal, and Olivia Todd for their careful review of this manuscript. The model was created using biorender.com.

## Author Contributions

**Conceptualization:** Jason R. Devlin, Judith Behnsen.

**Data curation:** Wenjing Peng.

**Formal analysis:** Jason R. Devlin, Wenjing Peng, Aiying Yu, Junyao Wang, Manmeet Singh, Peilin Jiang, Yehia Mechref, Judith Behnsen.

**Funding acquisition:** Yehia Mechref, Judith Behnsen.

**Investigation:** Jason R. Devlin, William Santus, Jorge Mendez, Wenjing Peng, Aiying Yu, Junyao Wang, Yehia Mechref, Judith Behnsen.

**Methodology:** Jason R. Devlin, Judith Behnsen.

**Project administration:** Yehia Mechref, Judith Behnsen.

**Resources:** Jason R. Devlin, Wenjing Peng, Xiomarie Alejandro-Navarreto, Kaitlyn Kiernan, Yehia Mechref, Judith Behnsen.

**Software:** Wenjing Peng, Yehia Mechref.

**Supervision:** Yehia Mechref, Judith Behnsen.

**Validation:** Jason R. Devlin.

**Visualization:** Jason R. Devlin, Wenjing Peng, Yehia Mechref, Judith Behnsen.

**Writing – original draft:** Jason R. Devlin, Judith Behnsen.

**Writing – review & editing:** Jason R. Devlin, William Santus, Wenjing Peng, Manmeet Singh, Yehia Mechref, Judith Behnsen.

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
