## [Decision Letter · Decision Letter 0]

1 Feb 2022

Dear Dr. Behnsen,

Thank you very much for submitting your manuscript "Salmonella enterica serovar Typhimurium chitinases modulate the intestinal glycome and promote small intestinal invasion." for consideration at PLOS Pathogens. As with all papers reviewed by the journal, your manuscript was reviewed by members of the editorial board and by several independent reviewers. The reviewers appreciated the attention to an important topic. Based on the reviews, we are likely to accept this manuscript for publication, providing that you modify the manuscript according to the review recommendations.

Sincerely,

Andreas J Baumler

Associate Editor

PLOS Pathogens

Brian Coombes

Section Editor

PLOS Pathogens

Kasturi Haldar

Editor-in-Chief

PLOS Pathogens

orcid.org/0000-0001-5065-158X

Michael Malim

Editor-in-Chief

PLOS Pathogens

orcid.org/0000-0002-7699-2064

Reviewer Comments (if any, and for reference):

Reviewer's Responses to Questions

**Part I - Summary**

Reviewer #1: In this study, Devlin et al investigate the role of chitinase-like glycoside hydrolases in the pathogenesis of Salmonella infection. Defined mutants lacking activity of two chitinase-like glycoside hydrolases, either as single mutants or in combination, exhibit decreased invasiveness and adhesion towards culture epithelial cells (IPEC-1 and T84 cell lines). Chitinase-deficient mutants are less invasive (about one order of magnitude) in the small intestine in a mouse model of Salmonella-induced colitis. ChiA is also required for efficient dissemination to systemic sites, but is dispensable for systemic replication. The defect of chitinase-deficient mutants in the mouse model is rescued by co-infection with the wild type, consistent with the fact that these chitinases are secreted. Evidence is presented that the glycome of cultured epithelial cells is changed upon Salmonella infection, and that this change is in part dependent on chitinase activity.

The main finding that Salmonella uses glycoside hydrolases to promote early invasion events in the small intestine is intriguing and novel, and should be of high interest to the audience of Plos Pathogens. The manuscript is well written. The majority of the conclusions are justified.

Reviewer #2: Manuscript by Devlin et al investigates the role of two annotated chitinases STM0018 (chiA) and STM0233 during Salmonella infection and pathogenesis. While ChiA was studied before, STM0233s role has not been investigated prior to this study. Authors demonstrate that both chitinases are upregulated in the mouse gut and facilitate epithelial cell adhesion and invasion. While wild type Salmonella that produces the two chitinases can hydrolyze the N-acetylglucosamine-containing glycans in the mouse gut and on epithelial cells, promoting the adhesion of the pathogen, mutants defective in one or both chitinases show reduced adhesion, invasion and dissemination to systemic organs. Finally the authors investigate the glycan profiles in wild type and the mutant bacteria to show how these chitinases increase the exposure of certain glycans including LewisX glycans so that the wild type Salmonella can increase its binding and facilitate virulence. Overall this is a well-performed study that increases of our understanding of initial steps of Salmonella pathogenesis and how this pathogen can bind and invade epithelial tissue. Study is supported by both in vitro and in vivo mouse experiments. There are few minor concerns.

Reviewer #3: This work relies on previous publications describing that chitin-degrading enzymes participate in bacterial pathogenesis. In this work, the authors address the role of two chitinases, in the pathogenesis of Salmonella Typhimurium. The work shows that chitinase mutants exhibit reduced adhesion to cultured intestinal epithelial cells, reduced invasion in the ileum and lower dissemination into systemic organs, compared to the wt strain. Chitinases also cause significant change into the glycan repertoire on intestinal epithelial cells, although some changes cannot be attributed to the glycolytic activity of the enzymes, and the underlying mechanism remains elusive.

The manuscript is well-organizes and the writing is clear and focused. The topic is very important and timely, and holds a potentially significant contribution to the field of pathogenesis. The experiments are well-performed and the data is compelling. My main concern is that the molecular underpinnings are still very obscure and it remains unclear how does the enzymatic activity of chitinases executes all the observed phenotype. However, the findings in this work are novel and important, and will probably provide a solid base for further studies.

**Part II – Major Issues: Key Experiments Required for Acceptance**

Reviewer #1: I only have one concern regarding the interpretation of the glycomics data and the conclusion that “STM0233 and ChiA enhance Salmonella adhesion to epithelial cells likely due to the exposure of mannose and increase in Lewis X binding residues.” This conclusion is primarily based on the experiment shown in Fig. 6. In this experiment, the authors have shown that Salmonella induces changes in the surface glycome of intestinal epithelial cells, but I am not convinced that STM0233 and ChiA are involved. If the process is driven by invading Salmonella bacteria, then the changes induced by STM0233 and ChiA could simply be due to differences in invasion (Fig. S6). I don’t think the entire glycomics survey should be repeated with an invasion-deficient strain or in the presence of cytochalasin D. However, one simple experiment would be to quantify one or two putative substrate on epithelial cells and rule out that invasion per se is a driver of the observed changes. At a minimum, this limitation could be discussed in the text.

Reviewer #2: 1- Throughout the study authors investigate the binding of Salmonella to ileum or colon using in vitro and in vivo studies. One would expect that reduced Salmonella adhesion, invasion and dissemination would be also reflected to reduced cytokine profiles in the tissues where the author see an effect of chitinases such as ileum. It is not clear why the authors they chose to use the cecal tissue to investigate the cytokine production. It is not suprising that they do not observe major changes in the cecal cytokine profiles at one time point except increased levels of il17a production in the animals infected by the chitinase mutants. It would be more informatory to study the cytokine profiles in the relevant ileal tissue or even peyers patches to show whether the chitinases effect on tissue invasion would reflect to the cytokine expression in these tissues.

2- Can the authors test the pef mutant in vitro to see whether pef fimbriae binds to the the exposed Lewis X glycans? This experiment would increase the impact of the paper if the authors can make that small but very important mechanistic link.

Reviewer #3: None

**Part III – Minor Issues: Editorial and Data Presentation Modifications**

Reviewer #1: Line 128: As written, it seems somewhat contradictory to me that the authors seek to investigate the role of chitinase activity during “Salmonella infection” and then assay transcription under laboratory conditions – maybe this could be reworded.

Fig. 1D and E: I am somewhat surprised that the invA mutant is recovered in lower numbers than the wild type. Would one expect that both strains are recovered in equal numbers after the cytochalasin D treatment since they both should adhere in a similar manner? It might be useful for the reader if the authors could discuss this potential caveat in the text.

Fig. 4: The y-axis is labelled as “arbitrary units” – aren’t these fold changes over untreated control mice?

Reviewer #2: The double mutant is only referred to as Δ2 only in Fig 4A through out the manuscript. It would be better for consistency to change it to ΔSTM0233 ΔchiA

If there is a more methodical way to show the glycan data to help the reader understand the different glycans, it would really help the manuscript. Not the clarity but it is difficult to digest this data for someone who is nor familiar with glycans. The little images by the graphs are a good idea though.

Reviewer #3: 1. Line 181 – Are the genes expression upregulated compared to expression in LB? If so, how does the expression of the normalizing gene change between LB an luminal content? I would suggest presenting the data as total copy numbers of mRNA (cDNA) per ml/gr/cfu, which would be a more informative.

2. Line 248 – change to Fig 2D

3. The authors discuss previous studies demonstrating only a minor contribution to chiA for invasion (lines 300-307). It should be worth mentioning in the text that one possible explanation could be redundancy, as the two chitinases investigated in the current work clearly present (Fig 1 & Fig 3), which might be overlooked in other studies.

PLOS authors have the option to publish the peer review history of their article (what does this mean?). If published, this will include your full peer review and any attached files.

Reviewer #1: No

Reviewer #2: No

Reviewer #3: No

Figure Files:

Data Requirements:

Reproducibility:

References:

---

## [Editor Report · Decision Letter 1]

23 Mar 2022

Dear Dr. Behnsen,

We are pleased to inform you that your manuscript 'Salmonella enterica serovar Typhimurium chitinases modulate the intestinal glycome and promote small intestinal invasion.' has been provisionally accepted for publication in PLOS Pathogens.

Best regards,

Andreas J Baumler

Associate Editor

PLOS Pathogens

Brian Coombes

Section Editor

PLOS Pathogens

Kasturi Haldar

Editor-in-Chief

PLOS Pathogens

orcid.org/0000-0001-5065-158X

Michael Malim

Editor-in-Chief

PLOS Pathogens

orcid.org/0000-0002-7699-2064
---

## [Editor Report · Acceptance letter]

6 Apr 2022

Dear Dr. Behnsen,

We are delighted to inform you that your manuscript, "Salmonella enterica serovar Typhimurium chitinases modulate the intestinal glycome and promote small intestinal invasion.," has been formally accepted for publication in PLOS Pathogens.

Best regards,

Kasturi Haldar

Editor-in-Chief

PLOS Pathogens

orcid.org/0000-0001-5065-158X

Michael Malim

Editor-in-Chief

PLOS Pathogens

orcid.org/0000-0002-7699-2064